# $H_2O$: Heavy-Hitter Oracle for Efficient Generative Inference of Large Language Models

**Zhenyu Zhang[1], Ying Sheng[2], Tianyi Zhou[3], Tianlong Chen[1], Lianmin Zheng[4], Ruisi Cai[1],**
**Zhao Song[5], Yuandong Tian[6], Christopher Ré[2], Clark Barrett[2], Zhangyang Wang[1], Beidi Chen[6,7]**
[1]University of Texas at Austin, [2]Stanford University, [3]University of California, San Diego,
[4]University of California, Berkeley, [5]Adobe Research, [6]Meta AI (FAIR), [7]Carnegie Mellon University
`{zhenyu.zhang,tianlong.chen,ruisi.cai,atlaswang}@utexas.edu, ying1123@stanford.edu,`
`{chrismre,barrett}@cs.stanford.edu, t8zhou@ucsd.edu, lianminzheng@gmail.com,`
`zsong@adobe.com, yuandong@meta.com, beidic@andrew.cmu.edu`

## Abstract

Large Language Models (LLMs), despite their recent impressive accomplishments, are notably cost-prohibitive to deploy, particularly for applications involving long-content generation, such as dialogue systems and story writing. Often, a large amount of transient state information, referred to as the KV cache, is stored in GPU memory in addition to model parameters, scaling linearly with the sequence length and batch size. In this paper, we introduce a novel approach for implementing the KV cache which significantly reduces its memory footprint. Our approach is based on the noteworthy observation that a small portion of tokens contributes most of the value when computing attention scores. We call these tokens *Heavy Hitters* ($H_2$). Through a comprehensive investigation, we find that ($i$) the emergence of $H_2$ is natural and strongly correlates with the frequent co-occurrence of tokens in the text, and ($ii$) removing them results in significant performance degradation. Based on these insights, we propose Heavy Hitter Oracle ($H_2O$), a KV cache eviction policy that dynamically retains a balance of recent and $H_2$ tokens. We formulate the KV cache eviction as a dynamic submodular problem and prove (under mild assumptions) a theoretical guarantee for our novel eviction algorithm which could help guide future work. We validate the accuracy of our algorithm with OPT, LLaMA, and GPT-NeoX across a wide range of tasks. Our implementation of $H_2O$ with 20% heavy hitters improves the throughput over three leading inference systems DeepSpeed Zero-Inference, Hugging Face Accelerate, and FlexGen by up to $29\times$, $29\times$, and $3\times$ on OPT-6.7B and OPT-30B. With the same batch size, $H_2O$ can reduce the latency by up to $1.9\times$. The code is available at `https://github.com/FMInference/H2O`.

## 1 Introduction

Large Language Models (LLMs) have demonstrated remarkable proficiency in a wide range of natural language processing applications such as content creation, summarization, and dialogue systems [1, 2, 3, 4]. However, their deployment is very costly. In addition to the widely-studied bottlenecks of model size and the quadratic cost of attention layers, the problem of the size of the KV cache, which stores the intermediate attention key and values during generation to avoid re-computation, is becoming increasingly prominent [5]. For instance, a 30 billion-parameter model with an input batch size of 128 and a sequence length of 1024 results in 180GB of KV cache. A natural approach is to limit its maximum size as is done in classical software or hardware caches [6]. However, it is challenging to reduce KV cache memory footprints in LLMs without accuracy drops.

37th Conference on Neural Information Processing Systems (NeurIPS 2023).

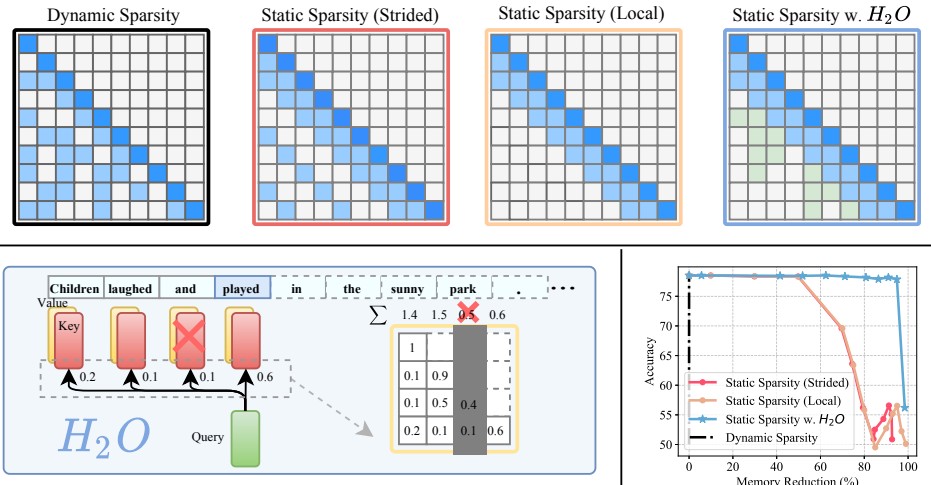

Figure 1: Upper plots illustrate symbolic plots of an attention map deploying different KV cache policies in LLM generation. Lower right: contrasts their accuracy-memory trade-off. Left: the overview of $H_2O$ framework.

While there exists substantial literature on sparse attention approximation in training, they have not seen wide adoption for alleviating KV cache bottleneck. First, most existing methods, e.g., Reformer [7] and Flash Attention [8], are designed to overcome the quadratic memory required by attention mechanisms when modeling long sequences but still require a *large cache size*. Second, variants like sparse transformer [9], low-rank based transformers [10, 11] or multi-query attention [12, 13, 5] can reduce the cache size, but directly applying them on pre-trained LLMs for generation results in *high miss rates* and degrades the accuracy as shown in Figure 1. Finally, some recent advances such as gisting tokens [14] can learn to compress the KV cache for documents, but their *expensive eviction policies* are difficult to deploy during generation.

Therefore, an ideal KV cache should have (i) a *small cache size* to reduce memory footprint, (ii) a *low miss rate* to maintain the performance and long-content generation ability of LLMs, and (iii) a *low-cost eviction policy* to reduce the wall-clock time during generation. However, there are three technical challenges. First, it is not immediately clear whether the size of the KV cache can be restricted—each decoding step might, in principle, require access to all previous attention keys and values. Second, identifying an optimal eviction policy that maintains generation accuracy is a combinatorial problem[1]. Finally, even if an optimal policy can be brute-forced, it is infeasible for deployment on real-world applications.

Fortunately, our preliminary exploration has yielded intriguing observations about the empirical properties of LLMs. These findings pave the way for the potential design of an efficient KV cache.

*Sparsity for small cache size*: We observe that even when trained densely, the attention matrices of LLMs are over 95% sparse at inference time (shown in Figure 2). This holds for a wide range of pre-trained LLMs. Therefore, only 5% of the KV cache is sufficient for decoding the same output token at each generation step, which suggests it may be possible to have up to a $20\times$ reduction in KV cache size without an accuracy drop.

*Heavy-Hitters for low miss rate*: We discover that the accumulated attention scores of all tokens in attention blocks adhere to a power-law distribution. It suggests that there exists a small set of influential tokens that are critical during generation, named heavy-hitters ($H_2$). $H_2$ provides an opportunity to step away from the combinatorial search problem and identify an eviction policy that maintains accuracy.

*Greedy algorithm for low-cost policy*: we surprisingly find that retaining the $H_2$ based on local statistics at each decoding step—summing the attention scores of only the preceding tokens—is as effective as considering the attention of future tokens (shown in Figure 2).

Based on the above, we first rigorously define the generative process of LLMs operating with a size-constrained KV cache in Section 2.1. Then we propose Heavy-Hitter Oracle ($H_2O$), a framework

---

[1]Belady's Algorithm is optimal for standard cache, but not necessarily for KV cache.

that exploits the properties of LLMs and uses simple, low-cost eviction policies that retrain the quality of LLMs throughout the generation process. Specifically,

- In Section 3, we explore the emergence of $H_2$ in attention, revealing their fundamental and critical roles: ($i$) $H_2$ exhibit a strong correlation of frequently co-occurring words in textual data; and ($ii$) removing $H_2$ completely damages the model's functionality. We demonstrate that $H_2$ can largely lower the cache miss rate of the existing policies mentioned above. Theoretically, assuming the attention scheme is submodular, $H_2$ corresponds to a greedy algorithm and is therefore near-optimal.
- In Section 4, we present a greedy but low-cost variant of $H_2$ which is dynamically determined by the accumulated attention score at each decoding step. We formulate the eviction policy with greedy $H_2$ as a variant of dynamic submodular maximization. The analysis shows that it results in a similar generative process as the one using the $H_2$ eviction policy.

We perform extensive experiments on OPT, LLaMA, and GPT-NeoX on a single NVIDIA A100 (80GB) GPU to evaluate $H_2O$ across a range of tasks from lm-eval-harness [15] and HELM [16]. We implement $H_2O$ on top of FlexGen that can easily adapt different cache eviction techniques to produce a system with high-throughput inference. Performance experiments show our framework achieves $29\times$, $29\times$, $3\times$ higher throughputs compared to three leading inference systems, DeepSpeed Zero-Inference [17], Hugging Face Accelerate [18], and FlexGen [19] respectively. With the same batch size, $H_2O$ achieves up to $1.9\times$ lower latency compare to FlexGen.

## 2    Related Work and Problem Setting

**Efficient Inference of LLMs.**    The substantial parameter counts of large language models (LLMs) present significant challenges for inference. To overcome this limitation, previous efforts have employed model compression techniques with specific designs to achieve efficient LLM inference, such as the method described in [20, 21, 22], which employs one-shot pruning on LLMs, resulting in negligible performance degradation even without retraining. Additionally, alternative approaches explore quantization methods specifically tailored to LLMs, as discussed in [23, 24, 25, 26, 27, 28]. Also, CoLT5 [29] employs a token-wise conditional computation strategy to reduce the overall computation cost. These methods address efficient inference from orthogonal perspectives and can be organically integrated. The techniques investigated in this study are closely associated with pruning or sparsity but focus on a distinct inference bottleneck, namely, KV cache. One closely related work[30] utilizes a learnable mechanism that determines necessary tokens during inference but requires an extra fine-tuning process, which makes it less practical.

**Sparse, Low-rank Attention Approx.**    The quadratic computational complexity of attention modules is one of the major bottlenecks of transformer inference [31]. Various efforts are devoted to addressing this challenge [7, 9, 10]. For example, Reformer [7] reduces the computational cost from quadratic to superlinear complexity via locality-sensitive hashing. Performer [10] employs positive orthogonal random features to approximate attention kernels. One relevant work, Sparse Transformer [9], introduces sparsity to reduce KV cache memory footprint and achieve an efficient attention mechanism, considered as our baseline in this paper. Moreover, SpAtten [32] utilizes accumulated attention scores to select important tokens for efficient attention inference while they don't consider the variance of token importance across attention heads and layers. Comparison with SpAtten is detailed in Appendix C.9.

**Caching.**    Caching, which plays a pivotal role in optimizing system performance, entails the development of effective eviction policies to handle frequently accessed data. Conventional approaches such as Least Recently Used and Least Frequently Used [33, 34] prioritize the recency and frequency of data access. And the design of KV cache encounters many similar challenges as traditional caching.

**LLM Inference Breakdown.**    The generative procedure of LLMs encompasses two distinct phases: (i) the *prompt* phase, in which an input sequence is utilized to produce the KV cache (consisting of the key and value embeddings), similar to the forward pass employed during LLM training; and (ii) the *token generation* phase, which leverages and updates the KV cache to generate new tokens incrementally. Each generation step relies on the previously generated tokens. The primary focus of this paper is to enhance the efficiency of the KV cache in attention during the token generation phase, thereby accelerating LLM inference.

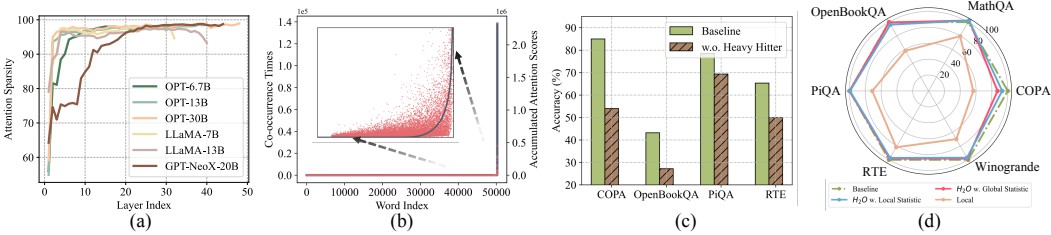

Figure 2: (a) Attention Sparsity in pre-trained LLMs. (b) The distribution of accumulated attention scores with respect to the corresponding word (red scatter) and the co-occurrence times of words in the data (gray curve). The x-axis represents the word index in the vocabulary. (c) The performance comparison between the baseline model with full KV and the model *w.o.* heavy hitter. (d) Comparison between the baseline model with full KV, H₂O with the local statistic, H₂O with the global statistic, and the model with only the most recent KV (Local). Apart from the baseline model, each model is evaluated with 20% KV cache budget.

## 2.1 Problem Formulation

We formally define the generative process with limited KV cache size. Denote attention query matrix as $Q \in \mathbb{R}^{n \times d}$ and key matrix as $K \in \mathbb{R}^{n \times d}$. $Q_{i,*}$ represents the $i$-th row of $Q$ and $K_{\leq i,*}$ represents the first $i$ rows of $K$. Let $k$ denote the budget of space and $k < n$. For simplicity, $K_{S_i,*}$ $(\in \mathbb{R}^{i \times d})$ denotes a sub-matrix of $K$ which selects $S_i$ rows from $K$. (For the non-selected rows $[i] \backslash S_i$, we put all zeros in that row) Eviction policy is defined as:

**Definition 2.1** (Eviction Policy, informal). *Let $S_{i-1}$ denote the source set. Let $S_i$ denote the target set. We defined the eviction policy $g : S_{i-1} \rightarrow S_i$ such that*

- $|S_i| = k$ *(KV cache size is not changing over the time)*

- $|S_i \backslash S_{i-1}| \leq 1$ *or equivalently $|S_i \cap S_{i-1}| \geq k - 1$ (we can evict at most 1 KV in the KV cache)*

Then, we define the generative process with our eviction policy.

**Definition 2.2** (The generative process with eviction policy, informal). *Let $k$ denote the size of the KV cache. For each $i \in [n]$, for the $i$-th token, we have*

- *Let $S_i \subset [n]$ denote the tokens in KV cache when predicting the $i$-th token.*

- *The information we have is a length-$i$ vector $o_i := D_i^{-1} \cdot \exp(Q_{i,*}(K_{S_i,*})^\top)$ (normalized attention)*

    - *scalar $D_i := (\exp(Q_{i,*}(K_{S_i,*})^\top) - 1_{[i] \backslash S_i}) \cdot \mathbf{1}_i$ (the evicted KV is set to 0, and we need to subtract them when computing the normalization)*
    - *Replacing $S_i$ by $[i]$ in the above definition of $o_i$ and $D_i$ leads to standard generative process.*

- *The eviction policy (Definition 2.1) updates $S_i$ based on $S_{i-1}$ and their corresponding information.*

**Remark 2.3.** *Our goal is to find a KV cache eviction policy such that the output of the generative process is similar or comparable to the original one without limiting the cache size.*

## 3 Observations

We present two key empirical insights of LLMs that inspire the design of H₂O, as follows.

### 3.1 Sparsity for Small Cache Size

Inspired by previous literature, which reveals the existence of attention sparsity in DistillBERT [35] and bounded-norm self-attention heads [36]. We first show an observation on the sparsity of attention in pre-trained LLMs. Then we discuss how it can potentially unlock the possibility of reducing KV cache size without an accuracy drop. Given the normalized attention score $\mathrm{Softmax}(QK^\top)$ matrix that is calculated by the query matrix $Q$ and the key matrix $K$, we set the threshold as one percent of the maximum value in each row and calculates the corresponding sparsity.

**Observation.** We conduct zero-shot inference with the pre-trained OPT model on the validation set of Wiki-Text-103. We plot the layer-wise sparsity within attention blocks and visualize the normalized attention score matrix. The results are presented in Figure 2 (a). We observe that although the LLMs are densely trained, the resulting attention score matrices are highly sparse, with a sparsity over 95% in almost all layers.

**Insights.** The attention blocks' sparsity suggests that access to all previous key and value embeddings is unnecessary for generating the next token. This suggests it is possible to evict unessential KV embeddings and reduce the requirement of KV cache during generation.

## 3.2 Heavy-Hitters for Low Miss Rate

The previous section showed the sparse nature of attention blocks in pre-trained LLMs, which provides the opportunity for designing small KV cache size while still maintaining the performance of LLMs. However, determining the best eviction policy that preserves generation accuracy presents a combinatorial challenge. Although Belady's Algorithm [37] is optimal and easy to compute for standard cache (offline), it is not applicable for KV cache design. Because once evicting important KVs, it could destroy the performance of LLMs due to the sequential dependency of LLM generation.

**Observation.** Fortunately, in the early stage of our exploration, we find that the accumulated attention scores of all the tokens within attention blocks follow a power-law distribution, as shown in Figure 2. This suggests the existence of a small set of tokens that are critical during generation. We denote those tokens as heavy-hitters ($H_2$). In order to verify the importance of these tokens, we compare the quality of LLM generation after masking heavy hitters with that of the original model. Not surprisingly, as shown in Figure 2, the accuracy drops drastically, confirming the importance of those tokens. Additionally, we can see the accumulated attention score of each word (in red dots) have a high correlation with their co-occurrences in the data (gray curve).

**Analysis.** First, based on $H_2$, we see an opportunity to side-step from the combinatorial search problem and design a KV cache eviction policy that preserves the LLM generation quality. We conduct an empirical study implementing a KV cache eviction policy that retains only the $H_2$ and the recent KV embeddings in the cache. The intuition is that recent words typically exhibit stronger correlations with current tokens. We assess the effectiveness of this eviction policy through pre-trained OPT-30B and six downstream tasks. The outcomes of these evaluations are illustrated in Figure 2. It is obvious that the $H_2$ based eviction policy can largely reduce the KV cache size without degrading the performance of OPT-30B.

Moreover, during the post analysis, inspired by [38], we find that $H_2$ based policy is related to the classical greedy algorithm (a polynomial-time algorithm with provable guarantees) under the assumption that the attention schema is submodular. We present details in Appendix D.

**Lemma 3.1** (informal). *Assuming the attention scheme is submodular, then greedily constructing the set $S_i$ (without cache size limitation) satisfies the near-optimal property in terms of submodular.*

# 4 Heavy-Hitter Oracle

The goal of this section is to propose the greedy algorithm using the $H_2$-based policy and to show the provable guarantees. We first present the $H_2$-based policy called $H_2O$ cache eviction policy and formulate its deployment in LLM generation as a variant of submodular maximization problem, named *dynamic submodular*. Then we present $H_2O$ in the generative process, followed by a practical example of deploying our proposal. Finally, we provide theoretical guarantees for $H_2O$ and show our efficient system implementation.

## 4.1 Greedy Algorithm for Low-Cost Policy

We have shown a simple yet effective KV cache policy based on $H_2$. However, it is impractical to deploy such an algorithm because we do not have access to the future-generated tokens. Fortunately, we empirically observe that local $H_2$, which is calculated using local statistics at every decoding step by summing up the attention scores of the previous tokens, is equally effective as taking into account the attention of future tokens (Figure 2). In the following, we formally define this dynamic attention score computation (with space limitation) as a novel dynamic submodular type problem.

**Definition 4.1** (Dynamic submodular framework, informal). *Define function $F : 2^{[n]} \times 2^{[n]} \to \mathbb{R}$, then for any set $Z \subset [n]$, we assume that $F(Z, \cdot) : 2^{[n]} \to \mathbb{R}$ is a submodular function w.r.t. to $Z$, i.e.,*

- *For all sets $X, Y \subset [n]$ satisfy that $Z \subset X \subset Y$,*

- *For all element $x \in [n]$ satisfy that $x \in [n] \backslash Y$,*

*we have $f(X \cup \{x\}) - f(X) \geq f(Y \cup \{x\}) - f(Y)$, where $f(\cdot) := F(Z, \cdot)$.*

**Remark 4.2.** *We provide practical insights of Definition 4.1. $X$ denotes the existing words in the* KV cache. *$Y$ is any superset of $X$. $x$ can be viewed as a "word" which is either newly added to* KV cache *or existing deleted from* KV cache. *An example $f$ can be attention score, i.e., see Algorithm 1.*

*If we load the sequence of $S_1, S_2, \cdots, S_n$ (we promise that $|S_i| \leq k$ and $|S_i \backslash S_{i-1}| \leq 1$) into Definition 4.1, i.e., for each $i \in [n]$, we choose $Z = S_i$, then it becomes a particular instance of the dynamic submodular problem.*

Next, we provide a formal description of our algorithm, followed by an example.

**Definition 4.3** (H$_2$O Eviction Policy). *Let $F_{\text{score}} : 2^{[n]} \to \mathbb{R}$ denote certain score function. Let $S_{i-1}$ denote the source set. Let $S_i$ denote the target set. We defined the eviction policy $g : S_{i-1} \to S_i$ s.t.*

- *$|S_i| = k$ (*KV cache *size is not changing over the time)*

- *$|S_i \backslash S_{i-1}| \leq 1$ or equivalently $|S_i \cap S_{i-1}| \geq k - 1$ (we can evict at most $1$* KV *in the* KV cache*)*

- *We construct $S_i \leftarrow (S_{i-1} \cup \{i\}) \backslash \{u\}$ as $u \leftarrow \arg\max_{v \in (S_{i-1} \cup \{i\})} F_{\text{score}}(S_{i-1} \cup \{i\} \backslash \{v\}\}$*

To describe our algorithm (Algorithm 1), we choose a particular instantiation of the function $F_{\text{score}}$, *i.e.*, the summation of that sets in the attention matrix.

---

**Algorithm 1** H$_2$ Eviction Algorithm

---

1: **procedure** H$_2$_EVICTION($Q, K \in \mathbb{R}^{n \times d}, k \in \mathbb{N}$)
2:     Let $k$ denote the budget size of cache
3:     $S_0 \leftarrow \emptyset$
4:     **for** $i = 1 \to n$ **do**
5:         **if** $i \leq k$ **then**
6:             $\overline{S}_i \leftarrow S_{i-1} \cup \{i\}$
7:         **else**
8:             $D_i \leftarrow (\exp(Q_{i,*}(K_{S_{i-1},*})^\top) - 1_{[i] \backslash S_{i-1}}) \cdot \mathbf{1}_i$
9:             $o_i \leftarrow D_i^{-1} \cdot (\exp(Q_{i,*}(K_{S_{i-1},*})^\top) - 1_{[i] \backslash S_{i-1}})$
10:            $F_{\text{score}}(T) := \sum_{s \in T} o_s$
11:            $G_i \leftarrow S_{i-1} \cup \{i\}$
12:            $u \leftarrow \underset{v \in G_i}{\arg\max} \, F_{\text{score}}(S_{i-1} \cup \{i\} \backslash \{v\}\}$
13:            $S_i \leftarrow (S_{i-1} \cup \{i\}) \backslash \{u\}$
14:         **end if**
15:     **end for**
16: **end procedure**

---

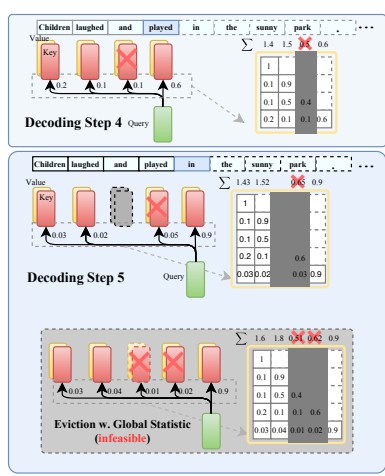

Figure 3: Illustration of Algorithm 1 during two consecutive decoding steps.

Figure 3 presents an illustrative example of our H$_2$ Eviction Algorithm. We assume that the budget size of KV cache is 3. Following the completion of the fourth decoding step, the KV embeddings associated with the third token are evicted based on the accumulated attention score. Consequently, these evicted KV embeddings become inaccessible in the subsequent decoding steps.

### 4.2 Theoretical Guarantee and System Implementation

We state a theoretical result as follows. The proofs and more details are provided in Appendix D.

**Theorem 4.4** (informal). *Under the mild assumption, let $k$ denote the budget of space limitation. If for each token, we greedily compute the attention score based on top-$k$ choice, then we can show the set $\widetilde{S}_i$ we generate each for token $i$ satisfy that $f(\widetilde{S}_i) \geq (1-\alpha)(1-1/e) \max_{|S|=k} f(S) - \beta$, where $\alpha, \beta > 0$ are parameters.*

**Remark 4.5.** *We remark the above theorem provides a theoretical explanation of why can we hope our greedy algorithm (with cache limitation) can provide a good solution to the problem.*

**Implementation Details.** We provide a general framework that can support any KV cache eviction algorithm and enhance throughput and reduce the latency of LLM generation with careful implementation. For example, to ensure I/O efficiency, we do not swap memory when stored KV is evicted, but directly fill with newly-added KV. More details are included in Appendix A.

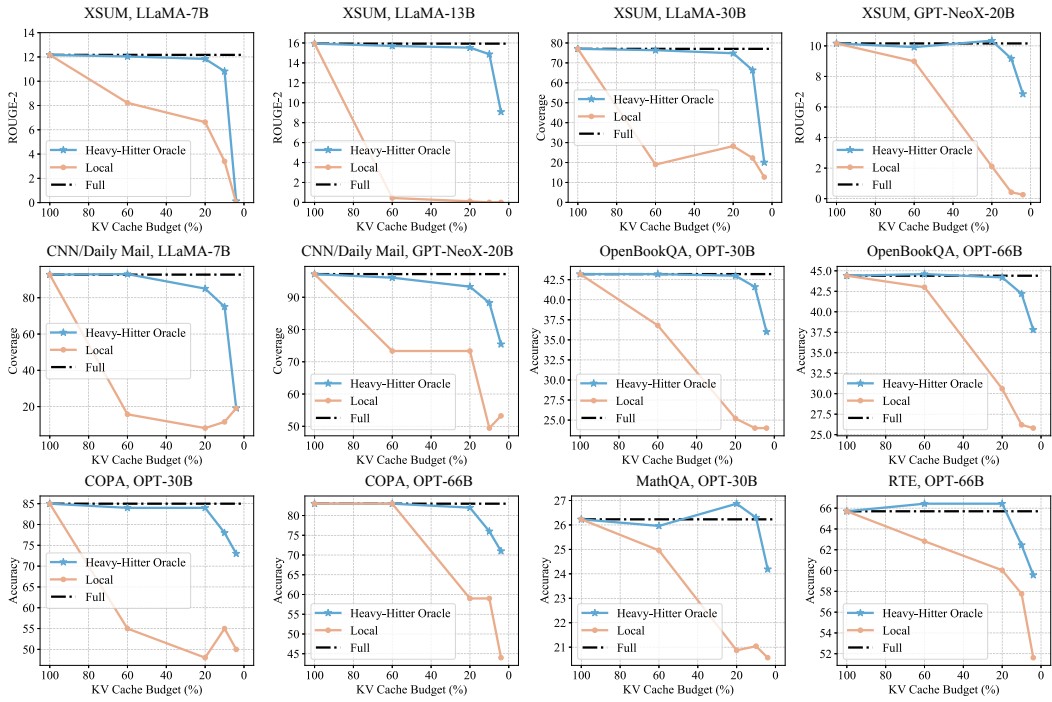

Figure 4: Comparsion results between the baseline model with full cache, our $H_2O$, and the "Local" strategy that utilizes the most recent KV embeddings.

## 5 Empirical Evaluation

In this section, our goal is to demonstrate that $H_2O$, a remarkably simple KV cache eviction policy is capable of enhancing end-to-end throughput and reducing latency in wall-clock while maintaining generation quality across a broad spectrum of domains and tasks.

- In Section 5.1, we show that $H_2O$ can reduce the memory footprint of KV cache by up to $5\times$ without accuracy degradation on a wide range of model architectures (OPT, LLaMA, GPT-NeoX), sizes (from 6.7B to 175B) and evaluation benchmarks (HELM and lm-eval-harness). More importantly, can enhance the performance of existing KV cache sparsification techniques.
- In Section 5.2, we demonstrate that $H_2O$ can increase the inference throughput by up to $3\times$, $29\times$, $29\times$ compared to the state-of-the-art inference engine FlexGen, DeepSpeed and the widely used Hugging Face Accelerate without compromising model quality.
- In Section 5.3, we present extensive ablation studies to show the effectiveness of $H_2O$ under different sequence lengths, especially the input with infinite sequence length and its compatibility with quantization.

All details (hyperparameters, data splits, etc.), along with additional experiments, are in Appendix A.

### 5.1 End-to-End Results

We demonstrate that $H_2O$ can reduce KV cache memory footprint by $5\text{-}10\times$ while achieving comparable accuracy on a majority of tasks.

**Setup.** Our experiments are based on three representative model families of LLMs, including the OPT [39] with model sizes, LLaMA [40], and GPT-NeoX-20B [41]. We sample eight tasks from two popular evaluation frameworks (HELM [16] and lm-eval-harness [15]): COPA [42], MathQA [43], OpenBookQA [44], PiQA [45], RTE [46], Winogrande [47], XSUM [48], CNN/Daily Mail [49]. Also, we evaluate our approach on recent generation benchmarks, AlpaceEval [50] and MT-bench [51], and the details are included in Appendix. We use NVIDIA A100 80GB GPU.

**Baselines.** Since $H_2O$ evenly assigns the caching budget to $H_2$ and the most recent KV, except for full KV cache, we consider the "Local" strategy as a baseline method. In addition, we also provide two different variants of Sparse Transformers (strided and fixed) as strong baselines. Also, the full

KV cache with fewer shots (0/1-shot) prompts are considered as the baseline, which has a similar sequence length of the 5-shot tasks with 20% KV cache budget.

**Main Results.** We evaluate LLMs with KV cache budget ranging from 4% to 100% on 5-shot downstream tasks. Results are summarized in Figure 4 and Table 1& 2. The following observations can be drawn: (1) With different KV cache budgets, our $H_2O$ demonstrates consistent and significant improvements against the "Local" strategy across various model sizes, model types, and downstream tasks. We can draw sim-

Table 1: Quantatively comparison between $H_2O$ with Full methods of different number of shots.

| Methods | PiQA | COPA | OpenbookQA | Winogrande |
|---------|------|------|------------|------------|
| Full | 80.09 | 81.00 | 44.80 | 71.51 |
| 0-shot Full | 78.89 | 76.00 | 41.40 | 70.00 |
| 1-shot Full | 79.11 | 76.00 | 43.60 | 70.24 |
| Local | 57.94 | 56.00 | 28.40 | 51.30 |
| $H_2O$ | 79.22 | 85.00 | 43.80 | 71.67 |

ilar conclusions comparing $H_2O$ with other baselines like Sparse Transformer; (2) Meanwhile, with less than 20% KV cache budget(*i.e.*, more than 5× memory reduction), $H_2O$ achieves comparable performance as the model with full KV embeddings; (3) $H_2O$ with 20% KV cache budget approximately uses 1.2 samples per input and show consistent improvement over zero-shot and one-shot full model that use 1 and 2 samples, respectively. (4) Our $H_2O$ shows consistent effectiveness in the more challenging long sequence generation tasks, XSUM, and CNN/Daily Mail.

**Analysis.** Since the evicted KV will not be seen in the future steps, dropping certain critical KV embeddings can cause a severe functional collapse, resulting in significant performance degradation, *e.g.*, in {LLaMA-13B, XSUM} {LLaMA-7B, CNN/Daily Mail}, the "Local" strategy collapses at 60% budgets while our $H_2O$ can still match the full cache performance with

Table 2: Results of different sparsification methods *w.* or *w.o.* $H_2$. Experiments are conducted with OPT-30B with 20% KV cache budget.

| Models | COPA | OpenBookQA | PiQA | Winogrande |
|--------|------|------------|------|------------|
| Full | 85.00 | 43.20 | 78.51 | 70.24 |
| Local *w.o.* $H_2$ | 48.00 | 25.20 | 55.82 | 49.17 |
| Local *w.* $H_2$ | 84.00 | 43.00 | 78.45 | 69.06 |
| Sparse Transformer (strided) *w.o.* $H_2$ | 50.00 | 24.60 | 56.20 | 47.59 |
| Sparse Transformer (strided) *w.* $H_2$ | 83.00 | 42.60 | 78.24 | 69.61 |
| Sparse Transformer (fixed) *w.o.* $H_2$ | 61.00 | 23.80 | 58.60 | 49.88 |
| Sparse Transformer (fixed) *w.* $H_2$ | 76.00 | 41.40 | 77.80 | 64.96 |

20% budgets. In some tasks, our methods even surpass the baseline models, which demonstrates a regularization effect of our $H_2O$. For example, in {OPT-66B, RTE}, {OPT-30B, MathQA} and {GPT-NeoX-20B, XSUM}, our $H_2O$ achieves an extra performance improvement of 0.73%, 0.64% and 0.18 with 20% KV cache budget, respectively. These consistent results validate the effectiveness of our $H_2O$ framework.

**Enhancing Baseline Techniques.** Importantly, we observe other sparsification baselines fail under an extremely low cache budget while combining the most recent KV embeddings with the ones of heavy hitters successfully achieves comparable performance as using full KV embeddings. From Table 2, we can observe that both "strided" and "fixed" sparse attention fail under 20% KV cache budgets, encountering a significant performance drop (up to 35% compared with the full cache). After combining with $H_2$, both approaches reach a similar performance as using full KV embeddings.

## 5.2 Heavy Hitter for High-Throughput Generative Inference

Table 3: Generation throughput (token/s) on a T4 GPU with different systems. In the sequence length row, we use "512 + 32" to denote a prompt length of 512 and a generation length of 32. "OOM" means out-of-memory. The gray text in the bracket denotes the effective batch size and the lowest level of the memory hierarchy that the system needs for offloading, where "C" means CPU and "G" means GPU.

| Seq. length | 512+32 | | 512+512 | | 512+1024 | |
|-------------|--------|------|---------|------|----------|------|
| Model size | 6.7B | 30B | 6.7B | 30B | 6.7B | 30B |
| Accelerate | 20.4 (2, G) | 0.6 (8, C) | 15.5 (1, G) | 0.6 (8, C) | 5.6 (16, C) | 0.6 (8, C) |
| DeepSpeed | 10.2 (16, C) | 0.6 (4, C) | 9.6 (16, C) | 0.6 (4, C) | 10.1 (16, C) | 0.6 (4, C) |
| FlexGen | 20.2 (2, G) | 8.1 (144, C) | 16.8 (1, G) | 8.5 (80, C) | 16.9 (1, G) | 7.1 (48, C) |
| $H_2O$ (20%) | 35.1 (4, G) | 12.7 (728, C) | 51.7 (4, G) | 18.83 (416, C) | 52.1 (4, G) | 13.82 (264, C) |

Table 5: Generation throughput and latency on an A100 GPU. In the sequence length row, we use "7000 + 1024" to denote a prompt length of 7000 and a generation length of 1024. "OOM" means out-of-memory.

| Seq. length | Model size | Batch size | Metric | FlexGen | $H_2O$ (20%) |
|---|---|---|---|---|---|
| 7000+1024 | 30B | 1 | latency (s) | 57.0 | 50.4 |
| 5000+5000 | 13B | 4 | latency (s) | 214.2 | 155.4 |
| 2048+2048 | 6.7B | 24 | latency (s) | 99.5 | 53.5 |
| 2048+2048 | 6.7B | 24 | throughput (token/s) | 494.1 | 918.9 |
| 2048+2048 | 6.7B | 64 | throughput (token/s) | OOM | 1161.0 |

We implement our KV cache eviction policy in a state-of-the-art inference engine, FlexGen [19], and report the throughput and latency improvements. $H_2O$ is orthogonal to existing optimizations in Flex-Gen, such as offloading and quantization, so they can be combined to achieve better performance.

**Setup** We conducted experiments on two GPUs: an NVIDIA T4 (16GB) GPU and an NVIDIA A100 (80GB) GPU. On the T4 GPU, we evaluate the generation throughput following the settings in the Flex-Gen paper. The evaluated models are OPT-6.7B and OPT-30B. When the model and KV cache do not fit into a single GPU, we turn on CPU offloading. The results of both pure GPU and GPU with CPU offloading are reported. All the speedup results are tested in an end-to-end setting, including both the pre-filling and generation phases. And it includes the time for constructing the $H_2O$ KV cache. We use synthetic datasets where all prompts are padded to the same length. The system is then required to generate the same number of tokens for each prompt. We test different combinations of prompt and generation lengths. We also test our method on real-world datasets (XSUM) for further assessment. The evaluation metric is generation throughput, which is the number of generated tokens / (prompt time + decoding time). We use DeepSpeed ZeRO-Inference [17], Hugging Face Accelerate [18], and FlexGen [19] as baselines. On the A100 GPU, with more GPU memory, we evaluate the performance of the systems with sequence lengths up to 10K. Although OPT is only trained on 2K sequence length, we benchmark the throughput and latency performance to show the potential of $H_2O$ for better models in the future.

Table 4: Results of generation throughput (token/s) on a T4 GPU with different systems on real-world datasets, XSUM.

| Model size | 6.7B | 30B |
|---|---|---|
| Accelerate | 11.98 (1, G) | 0.23 (2, C) |
| DeepSpeed | 3.52 (6, C) | 0.31 (2, C) |
| FlexGen | 10.80 (1, G) | 3.29 (44, C) |
| $H_2O$ (20%) | 30.40 (1, G) | 6.70 (180, C) |

**Results.** Table 3& 4 shows the generation throughput of all systems on the T4 GPU. With our KV cache eviction policy, the memory usage is reduced, which brings two advantages: 1) we can use a much larger batch size; 2) we can make a setting from requiring offloading to not requiring offloading. As shown in Table 3& 4, $H_2O$ with a 20% budget improves the generation throughput over FlexGen, DeepSpeed, and Accelerate by up to $3\times$, $29\times$, and $29\times$, respectively, across both synthetic and real-world dataset. The results on the A100 GPU with sequence lengths from 4K to 10K are listed in Table 5. With the same batch size, $H_2O$ can reduce the latency by $1.1 - 1.9\times$ compared to FlexGen. Additionally, $H_2O$ saves memory so it allows a larger batch size, which brings $2.3\times$ improvement on generation throughput for OPT-6.7B.

## 5.3 Ablation Results

We present extensive ablation studies of $H_2O$ on (1) infinite-length input, (2) different number of shots, (3) compatibility with quantization methods on KV cache, and (4) dissecting the effectiveness of different components. We find a surprising property of $H_2O$ – it not only improves the efficiency of LLMs, but also increases the diversity of the generated text.

*Q1:* **Can $H_2O$ empower LLMs to process infinite-length inputs?** *A1:* **Effective generation with sequence length up to four million tokens.** Some recent works [52, 53] demonstrate the possibility of handling infinite-length inputs, a notable challenge in current LLMs. These methods employ an attention sink that retains the first few tokens and applies position rolling in the KV cache, empowering LLMs to process infinite-length inputs. Inspired by this progress, we further implement our $H_2O$ for infinite-length inputs. Figure 5 showcases the positive results of $H_2O$, *i.e.*, $H_2O$ can empower LLMs to tackle input with length up to four mil-

lion tokens, achieving a better performance (lower perplexity) than the original StreamLLM method [52] across various cache size. Further comparisons are reported in Appendix C.4.

*Q2:* **Does the number of shots during inference effects the effectiveness of** $H_2O$**?** *A2:* **Effective across zero-shot to ten-shots inference.** We further examine $H_2O$ under different numbers of shots during inference, and the results are reported in Table 10 and Figure 8. With different shots inference, our $H_2O$ achieves matching performance (difference less than $1.00\%$) as the full model across different downstream tasks. The "Local" strategy encounters significant performance degradation (up to $37.00\%$. Such results demonstrate the effectiveness of our $H_2O$ under different inference scenarios. More details about zero-shot and one-shot inference are reported in Appendix C.3.

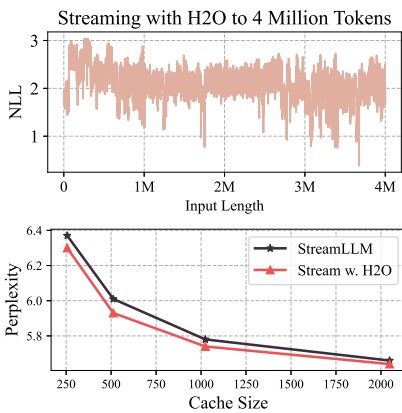

Figure 5: (Upper) streaming with $H_2O$ to handle inputs with sequence lengths of four million tokens. (Bottom) Perplexity comparison between the original StreamLLM method and our $H_2O$, results are collected on the first text sample of PG-19 [54].

*Q3:* **Compatible with Quatization?** *A3:* **Yes.** To pursue further efficiency, we show the compatibility of $H_2O$ with another orthogonal approach, *i.e.*, quantization in Table 6. We use OPT-30B as our base model and COPA, OpenBookWA, and PiQA as evaluation tasks. Intuitively sparsity and quantization are highly related so combining them might introduce larger errors. Surprisingly the combination almost always achieves better accuracy than $H_2O$ or quantization alone. Experiments about throughput improvement are detailed in Appendix C.2.

*Q4:* **When does** $H_2O$ **match the baseline with full** KV **embeddings?** *A4:* **With both** $H_2$ **and the recent tokens.** We investigate the separate effects of KV embeddings of $H_2$ and the local tokens. We conduct experiments on 4 tasks with OPT-13B and OPT-30B. For each task, we compare the performance of three KV cache eviction policies, including only the KV embeddings of $H_2$, only the ones of local tokens, and our $H_2O$ that keep both. As shown in Table 9, only retaining the embeddings of $H_2$ or local tokens can't maintain a similar performance as the model using full embeddings, with a performance degradation from $2.85\%$ to $22.75\%$. Incorporating both components, our $H_2O$ successfully retains the baseline performance with full embeddings. Besides, the model with only $H_2$ shows a consistent improvement against the one with only local tokens, which indicates $H_2$ might contribute more to maintaining the performance.

*Q5:* **Extra benefits from** $H_2O$**?** *A5:* **Increased diversity of generated text.** Besides all the benefits of our $H_2O$, we also observe an bonus introduced by $H_2O$, *i.e.*, the improved diversity of generated content. The results are reported in Appendix C.1. Given the same prompts, we visualize the generated text of the models with different KV cache budgets. Compared with the model of full KV cache, our $H_2O$ can generate sentences with fewer repeated words and more creativity.

## 6 Conclusion and Discussion

In this paper, we study one of the key bottlenecks of LLM deployment, KV cache, particularly for long-content and large-batch generation applications. We propose $H_2O$, a simple KV cache eviction policy for significantly reducing its memory footprint. The main insight of our approach is the recognition of a subset of tokens, known as Heavy Hitters, which contribute the most value when computing attention scores. We formulate the KV cache eviction as a dynamic submodular problem and provide the theoretical guarantees for our algorithm. Through extensive evaluations, we demonstrate that $H_2O$ can significantly improve end-to-end throughput and decrease latency in wall-clock time, without compromising the generation quality of LLMs across a variety of tasks.

## 7 Acknowledgement

Ying Sheng and Clark Barrett are partly supported by NSF-2110397 and the Stanford Center for Automated Reasoning. Z. Wang is in part supported by a Google Research Scholar Award and the NSF AI Institute for Foundations of Machine Learning (IFML).

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

# Appendix

## Table of Contents

# A   More Implementation Details

In this section, our goal is to provide the details of system implementation (mentioned in Section 4.2) and experiment settings (mentioned in Section 5), as well as the pseudocode.

**System Details.**   We implement $H_2O$ on top of FlexGen. FlexGen is a white-box implementation of OPT models, and we have done some surgery on handling the KV cache. Specifically, for the given parameter $K$, we always maintain a list of KV cache with the first $K$ entries as heavy hitters, and the last $K$ entries as most recent tokens. In order to avoid data movement in memory, the memory for KV cache is preallocated. We use a circular queue to update the last $K$ entries efficiently.

**Experiment Details.**   Our study involves the evaluation with varying sizes of KV cache that encompass $4\%$, $10\%$, $20\%$, and $60\%$ of the prompt's length. We select two tasks (XSUM and CNN/Daily Mail) from the HELM framework [16] and present the performance based on $1000$ test samples. Additionally, we employ the lm-eval-harness framework [15] for six other tasks (COPA, MathQA, OpenBookQA, PiQA, RTE, and Winogrande). For the tasks derived from the HELM framework, we report the performance for zero-shot inference, while for the tasks from lm-eval-harness, we default to conduct five-shot inference.

**Pseudocode.**   We show a simplified pseudocode below to demonstrate our implementation skeleton. The function `generation_loop()` is the base loop in FlexGen that controls prefetch and overlap the I/O streams and computation. Then in function `compute_layer()`, the function `attention_forward()` will be called for attention layers. During prefill iterations, the function `compute_attention()` will return $K$ heavy hitters and $K$ recent tokens if the prompt has a length greater than or equal to $2K$, otherwise return all the KV cache. The function `compute_attention()` will return new KV cache and the indices for evicted entries during decoding iterations, which would be the place to implement a customized eviction strategy. (If the current number of stored KV cache is less than $2K$, the eviction will be ignored.) During each decoding iteration, the oldest one among the last $K$ tokens (if we have no less than $K$ tokens' KV cache stored) will be removed from the reserved last $K$ entries, one heavy hitter will be evicted for each head, and the newest token will be added to the KV cache with a position in the last $K$ entries. This happens in the function `store_cache()`.

```python
def generation_loop(...):
  # Prologue
  ...
  # Generate
  for i in range(gen_len):
    for j in range(num_layers):
      for k in range(num_gpu_batches):
        load_weight(i, j+1, k)
        load_cache(i, j, k+1)
        store_hidden(i, j, k-1)
        load_hidden(i, j, k+1)
        compute_layer(i, j, k)
        store_cache(i, j, k-1)
        sync()
  # Epilogue
  ...

# h is the hidden states (activations)
def attention_forward(h, ...):
  # the read/write buffer are intermediate stops for prefetching
  if prefill:
    h, new_k_cache, new_v_cache = compute_attention(h, ...)
    # select K heavy hitters and K recent tokens
    new_k_cache, new_v_cache = select(new_k_cache, new_v_cache, K)
    cache_write_buffer.store(new_k_cache, new_v_cache)
  else:
    k_cache, v_cache = cache_read_buf.pop()
    # evict_ids track the entries that will be evicted
    h, new_k_cache, new_v_cache, evict_ids =
            compute_attention(h, k_cache, v_cache, ...)
```

```
    cache_write_buffer.store(new_k_cache, new_v_cache, evict_ids)
  return h

def store_cache(...):
  if prefill:
    # store cache directly
    ...
  else:
    k_new, v_new, evict_ids = cache_write_buffer.pop()
    # circular queue for the last K entries
    # extract the index for the oldest token at i-th iteration
    oldest = ((i - 1) % K) - K
    # update the KV cache (k_home and v_home)
    cache_replace(k_home, evict_ids, k_new, K, oldest)
    cache_replace(v_home, evict_ids, v_new, K, oldest)
```

# B  Extended Related Works, Discussions, and Limitations

The goal of this section is to first introduce more background and related works for Section 2, then describe some previous attempts in our experiments as well as discuss the social impact and limitations of this work.

## B.1  Extended Related Works

**Quantization, Pruning, Distillation for Inference.**   Previously, model compression algorithms have been extensively investigated as a viable approach for mitigating the computational resource requirements of model inference. These algorithms can be broadly categorized into three groups: (1) quantization [55, 56, 57, 58], which involves mapping model parameters or activations from high-precision data types to low-precision counterparts, such as using 8-bit integers instead of the commonly employed 32-bit floating point format; (2) pruning or sparsity [59, 60, 61, 62], which aims to eliminate unnecessary neurons or weights within the models; (3) and distillation [63, 64, 65, 66] where predictions from larger models are utilized as supervised information to train smaller models.

**Transformer in NLP.**   Transformers [67] as a popular option have been frequently adopted by plenty of natural language processing (NLP) applications with prevailing successes [68, 69, 70, 71, 72, 46, 73, 13, 74, 75]. Roughly, modern transformer-based networks can be categorized into two groups: (1) Encoder-Decoder or Encoder-only (*i.e.*, BERT-style models [76]). This type of transformers commonly leverages the Masked Language Modeling task which encourages models to capture the intrinsic relationship between words and their context. Notable examples include BERT [76], RoBBERTa [69] and T5 [77]. (2) Decoder-only (*i.e.*, GPT-style models [78]). Usually, this group of transformers adopts the Casual Language Modeling task, which is optimized to generate the next word/token in a sequence based on the preceding words/tokens. Such an autoregressive manner is highly preferred by downstream tasks like text generation and question answering. GPT-3 [79], OPT [39], PaLM [13], and BLOOM [80] are representative architectures within this huge family.

**Training of Transformer.**   Training a gigantic transformer-based model is not trivial. It notoriously suffers from various issues such as overfitting, instability, etc. [81, 82, 83] analyze these bottlenecks from the optimization perspective. To address the issues, a great amount of pioneering effort is devoted, including data augmentations [84, 85], a better initialization [86, 83, 87, 88], customized optimizers [89], improved normalization [90, 91], weight decay [92], and early stopping. However, there is still a long way to go before we can fully clarify the mystery of transformer training.

## B.2  Discussions and Limitations

**Previous Attempts.**   During our experiments, we find several noteworthy observations. In $H_2O$, employing the accumulated attention score to evict KV embeddings can lead to a potential bias favoring the least recent tokens. This bias arises because most previous tokens have a higher number of attention scores, resulting in a higher accumulated attention score and, consequently, a greater likelihood of being retained. To address this concern, we conducted an additional experiment utilizing

the averaged attention score to determine which KV embeddings should be retained. However, this alternative approach resulted in performance degradation. Additionally, we observed a significant proportion of $H_2$ occurrences at the beginning of sentences. This finding suggests that the initial tokens play a substantial role in subsequent generation tasks.

**Social Impact.** Our work represents an initial effort in designing a KV Cache policy, a realm that has been relatively unexplored and yet is a significant bottleneck in LLMs. The proposed Heavy Hitter Oracle ($H_2O$) provide a solution to improve the efficiency of LLM generation, which can save energy cost and contribute to green AI. Besides, our approach also serves as a source of inspiration for future advanced algorithm designs. We envision $H_2O$ as a foundational framework that could facilitate further innovation in this area. Moreover, long content generation is an area of growing importance that currently grapples with several efficiency issues. We hope our work that supports the generation of very long sequences will support further research in this direction, particularly in terms of enhancing consistency, devising superior evaluation methods, and establishing robust benchmarks.

Furthermore, another contribution of this study is the formulation of a dynamic submodular framework. We believe that this theoretical framework possesses the potential to be applicable beyond specific domains of interest. For instance, there may exist numerous other dynamic problems where the task involves solving a submodular problem with slight variations at each time.

**Limitations.** Furthermore, despite the notable advancements in throughput of our $H_2O$, implementing LLMs for generative inference remains challenging due to the immense parameter count. As a substantial portion of these parameters is encompassed within the MLP blocks, building upon our observations of $H_2$ occurrences in the MLP blocks, future research efforts can be directed towards leveraging the characteristics of $H_2$ to devise an offloading policy. Such a policy can potentially enhance the efficiency of LLM inference even further.

# C  Extended Experiments

In this section, our goal is to demonstrate that $H_2O$ can improve the diversity of generated text (mentioned in Section 5.3), throughput is further improved when combined with quantization (mentioned in Section 5.3), and superior performance to handle extremely long length inputs (up to four middle tokens, mentioned in Section 5.3). Moreover, additional investigations about $H_2$ are reported, including experiments under zero-shot/one-shot inference regime; $H_2O$ can also enhance the "Top-K" baseline; extra results of $H_2$ in attention blocks; the emergence of $H_2$ in MLP blocks as well as its properties.

## C.1  Increased Diversity of Generated Text

> **Model Input**
>
> In the year 2087, humanity has achieved remarkable technological advancements and established colonies on multiple planets within the Milky Way galaxy. Interstellar travel has become commonplace, with faster-than-light spacecraft enabling people to explore distant star systems. Earth has undergone significant changes due to sustainable development efforts, such as harnessing renewable energy sources and implementing widespread ecological restoration projects. However, alongside these triumphs, new challenges have emerged, including the rise of artificial intelligence, ethical dilemmas surrounding genetic engineering, and interplanetary political tensions. Against this backdrop, a team of intrepid scientists embarks on a mission to uncover the secrets of an ancient alien civilization, hidden deep within an uncharted exoplanet. As they navigate treacherous terrains and encounter otherworldly phenomena, they must confront their own fears and reconcile humanity's thirst for knowledge with the potential consequences of uncovering secrets that were better left buried. The fate of both their mission and the future of humanity hang in the balance.

> **OPT-6.7B Full Cache Output**
>
> The game is set in 2087, a few years after the events of the first game. The player takes the role of a team of scientists who are sent to an uncharted planet to investigate a mysterious signal. The team is composed of a pilot, a scientist, a mechanic, and a technician.

> **OPT-6.7B Local 20% Cache Output**
>
> ·············································,,,,,,,,,,,,,,,,,,,,,,,,,

> **OPT-6.7B $H_2O$ 20% Cache Output**
>
> The game is set in the year 2087, and is a first-person exploration game. The player takes on the role of a scientist who has been sent to a distant planet to investigate a mysterious signal. The planet is inhabited by a race of intelligent beings known as the "Titans," who have been

Figure 6: Visualization of one generation example with OPT-6.7B. Results are compared between the baseline model with full cache, our $H_2O$, and the "Local" strategy that utilizes the most recent KV embeddings.

Given the same prompt text, we visualize the generated text with OPT-6.7B and LLaMA-7B across different methods, including the baseline model with full cache, our $H_2O$, and the "Local" strategy. Results are reported in Figure 6 and 7. Even with less KV cache budget, our $H_2O$ can generate more diverse content. Specifically, with the OPT-6.7B, the full model generates some redundant works, like "a few years after the events of the first game" while our $H_2O$ describes "the game is a first-person exploration game". As a comparison, when all KV cache budget is assigned to the most recent tokens, the model fails to generate meaningful text and only repeats the word "." and ",". Similar observations can also be drawn from the results of LLaMA-7B, in which the full model repeatedly says "so moving that", "so moved that", and "began to cry" while our $H_2O$ describes both the people and the environment.

Moreover, We conducted a quantitative comparison via diversity metric(Self-BELU [93]). We randomly sample 100 instances from the XSUM dataset, as prompts. Subsequently, we employed the LLaMa-7B to generate text of equal length. The full model reaches a Self-BELU score of 0.0057 while $H_2O$ and the local method achieve 0.0051 and 0.0436, respectively. The comparatively lower Self-BELU score of $H_2O$ indicates the slightly increased diversity of generated text.

| Model Input |
| --- |
| In a small, bustling cafe nestled in the heart of a vibrant city, a serendipitous event unfolded, leaving a lasting impression on all who witnessed it. As the patrons sat sipping their coffees and engaging in animated conversations, a talented street musician entered the cafe, carrying a weathered guitar and radiating an aura of creativity. |

| LLaMA-7B Full Cache Output |
| --- |
| He began to play, and the patrons were captivated. The musician's performance was so moving that the patrons began to applaud, and the musician was so moved that he began to cry. The patrons were so moved that they began to cry, and the musician was so |

| LLaMA-7B Local 20% Cache Output |
| --- |
| He ( ( ( ( ( ( ( ( ( ( ( ( ( ( ( (, [)), 1999, 2000, 2001, 1, and, and, and, and, and, and, and, and, and, and, and, |

| LLaMA-7B $H_2O$ 20% Cache Output |
| --- |
| He began to play, and the room was filled with the sound of his music. The patrons of the cafe were enthralled by the music, and the atmosphere was electric. The cafe was packed with people, all of whom were enjoying the music. The musician was a young |

Figure 7: Visualization of one generation example with LLaMA-7B. Results are compared between the baseline model with full cache, our $H_2O$, and the "Local" strategy that utilizes the most recent KV embeddings.

Table 6: Compatibility with Quantization.

| Models | COPA | OpenBookQA | PiQA |
| --- | --- | --- | --- |
| Full | 85.00 | 43.20 | 78.51 |
| $H_2O$ | 84.00 | 43.00 | 78.45 |
| Quant-4bit | 84.00 | 43.28 | 78.67 |
| $H_2O$ *w.* Quant-4bit | 84.00 | 43.20 | 78.80 |

## C.2 Throughput Improvement of $H_2O$ Combining with Quantization

Table 6 demonstrates $H_2O$ are capable of quantization and perverse the full model's performance with even 4-bit quantization. To further explore the compatibility of our $H_2O$ with quantization with respect to throughput improvement, we conduct an additional evaluation with the quantization method implemented in FlexGen (Note that [94] employed a different 4-bit quantization method). The corresponding results are presented in Table 7. Notably, for OPT-6.7B, we observed extra performance enhancements in $H_2O$ when utilizing quantization compared to the vanilla version. This improvement results from the GPU memory freed by weight quantization, which allows for a significant increase in batch size. However, it should be emphasized that the quantization method employed in FlexGen is not implemented most efficiently, resulting in considerable computational overhead. Despite the batch size being enlarged by 20 times, the actual throughput improvement is less than 2 times. Nevertheless, it is important to acknowledge the potential benefits of combining $H_2O$ with quantization, as exemplified by the ability to increase the batch size further. For instance, the implementation of 4-bit quantization could be accelerated by an optimized CUDA kernel.

Table 7: Generation throughput (token/s) with different systems without offloading. We use $H_2O$-c denotes the $H_2O$ with 4-bit weights compression. In the sequence length row, we use "512 + 32" to denote a prompt length of 512 and a generation length of 32. "OOM" means out-of-memory. The gray text in the bracket denotes the batch size. We run OPT-6.7B on a single T4 GPU.

| Seq. length
Model size | 512+32
6.7B | 512+512
6.7B | 512+1024
6.7B |
| --- | --- | --- | --- |
| Accelerate | 20.4 (2) | 15.5 (1) | OOM |
| DeepSpeed | OOM | OOM | OOM |
| FlexGen | 20.2 (2) | 16.8 (1) | 16.9 (1) |
| $H_2O$ (20%) | 35.1 (4) | 51.7 (4) | 52.1 (4) |
| $H_2O$-c (20%) | 50.5 (70) | 72.5 (52) | 62.3 (44) |

## C.3 Effectiveness on Zero-shot and One-shot Inference

When facing zero-shot and one-shot inference tasks, $H_2O$ can successfully mitigate the memory requirements of the KV cache by up to $5\times$ under both one/zero-shot inference across different tasks, achieving matching performance as the model with full KV cache while the local method fails. However, unlike 5-shot, we also found that some tasks are more difficult, and the model requires more information to generate the correct content, resulting in a higher KV cache budget (30-40%). This might be expected since 0/1 shots in our benchmarks have shorter sequence lengths ranging from 100 to 300.

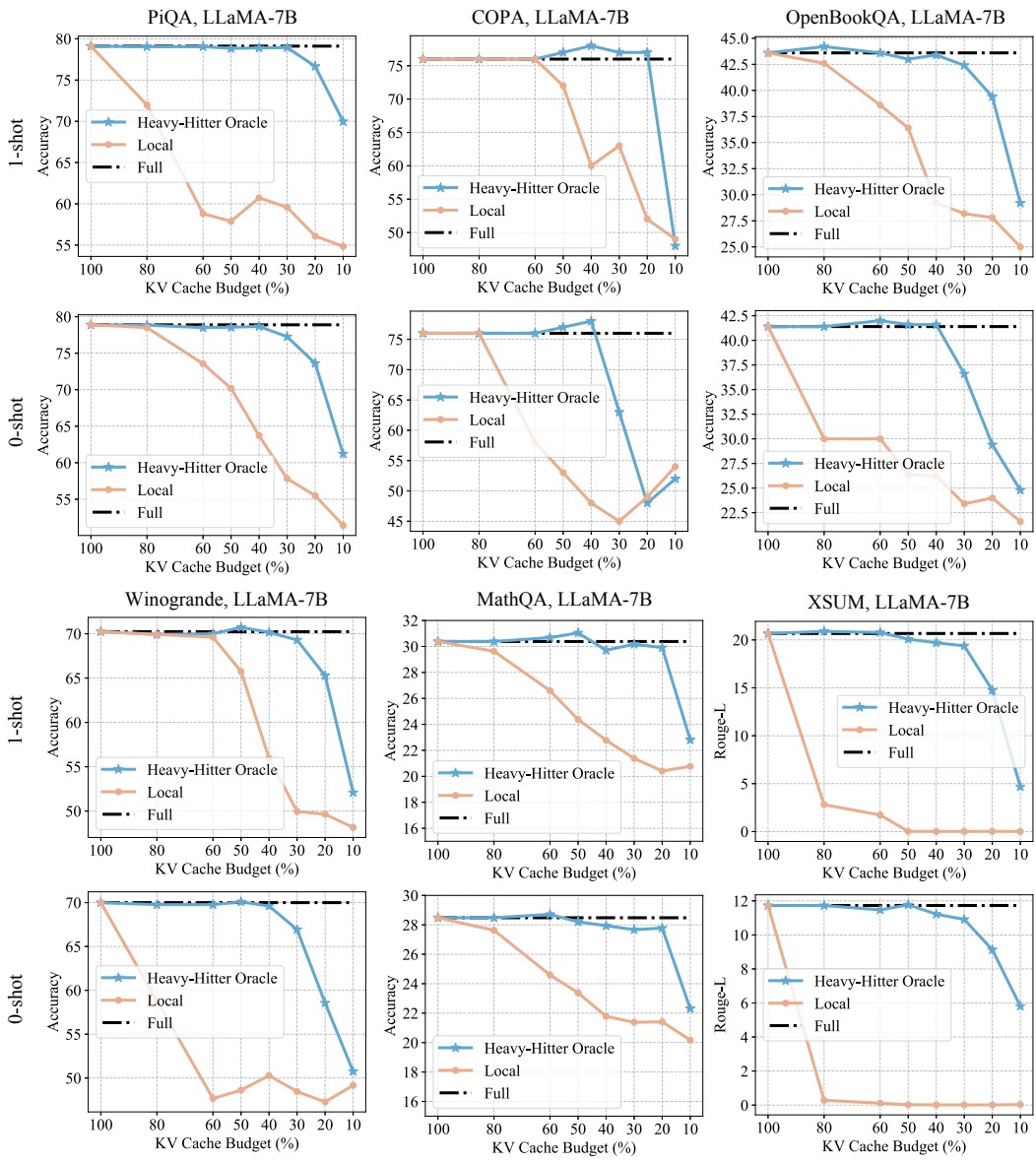

Figure 8: Comparsion results on zero-shot and one-shot inference.

## C.4 Comparison with StreamingLLM

Recent works, such as StreamLLM [52] and LM-Infinite [53], have shown promising potential in enabling Language Models (LLMs) to handle input of infinite length. They achieve this by only retaining the initial tokens and a limited local context. However, this approach may pose challenges for certain tasks where vital information lies within the middle of the input and would be lost using this strategy. We investigate it through two specific types of tasks:

**Multi-document question answering** [95]: In this task, each test sample comprises ten documents, followed by a question. The key information to answer this question is stored in one of the documents. We rearranged the crucial document's position and found that the eviction strategy in StreamLLM or LM-Infinite can not perform well when the key document has been dropped.

**Text Summarization Task** (XSUM and CNN-DailyMail): Text summarization tasks require models to generate concise summaries of lengthy texts. Effective summarization demands a comprehensive understanding of the entire document, making it challenging when crucial information is dispersed throughout the input. In particular, summarization often relies on long-context attention, and critical information may not be effectively captured within the limited local tokens.

The results are reported in Figure 9, illustrating a consistent decline in the performance of StreamLLM. Since StreamLLM will always maintain the first few tokens as well as the local tokens, regardless of various input content, such a strategy will inevitably result in the loss of crucial information and subsequently lead to a decrease in performance. In contrast, our $H_2O$ delivers markedly superior performance. Also, with $H_2O$, The model can successfully stream to four million tokens.

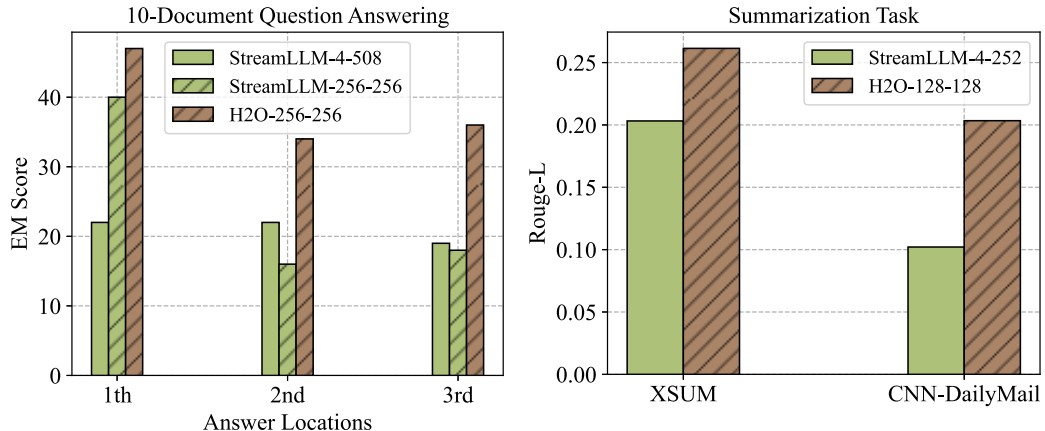

Figure 9: Comparison results of StreamLLM [52] and our $H_2O$ on generization tasks. The number in each method represents the KV Cache budget of the start/heavy-hitter tokens and the local tokens, respectively. For example, H2O-256-256 means maintaining 256 Heavy-Hitters and 256 local tokens.

## C.5 Enhancing the "Top-K" Baseline

We find $H_2$ can further enhance another strong baseline with a "Top-K" strategy. The results are reported in Table 8. After combing with $H_2$, the "Top-K" method achieves an extra improvement with up to $2.00\%$ accuracy across 4 different tasks.

Table 8: Results of the "Top-K" method *w.* or *w.o.* $H_2$. Experiments are conducted with OPT-30B with $20\%$ KV cache budget.

| Models | COPA | OpenBookQA | PiQA | Winogrande |
|---|---|---|---|---|
| Full | 85.00 | 43.20 | 78.51 | 70.24 |
| TopK *w.o.* $H_2$ | 80.00 | 41.40 | 76.96 | 65.35 |
| TopK *w.* $H_2$ | 82.00 | 42.80 | 77.96 | 66.48 |

## C.6 Separate Effects of Heavy-Hitter and Local Tokens

Table 9 demonstrates the separate effects of Heavy-Hitters and the local tokens. And we can observe Heavy-Hitters contribute more to maintaining the performance of models.

## C.7 Performance of Inference with Different Number of Shots.

Table 10 demonstrates our $H_2O$ achieves consistent improvements across different number of shots during inference and maintains the full model's performance with $20\%$ memory budget.

Table 9: Ablation study of $H_2O$ across different tasks.

| Tasks | Models | Full | w. Local | w. $H_2$ | w. Local + $H_2$ |
|---|---|---|---|---|---|
| PiQA | OPT-13B | 77.37 | 54.62 | 76.12 | 77.26 |
| | OPT-30B | 78.51 | 55.82 | 67.25 | 78.45 |
| OpenBookQA | OPT-13B | 41.40 | 25.60 | 30.40 | 41.20 |
| | OPT-30B | 43.20 | 25.20 | 26.60 | 43.00 |
| MathQA | OPT-13B | 26.67 | 22.04 | 23.82 | 26.93 |
| | OPT-30B | 26.23 | 20.87 | 21.98 | 26.87 |
| Winogrande | OPT-13B | 68.59 | 49.96 | 51.85 | 67.32 |
| | OPT-30B | 70.24 | 49.17 | 47.36 | 69.06 |

Table 10: Results under different sequence length of OPT-30B with $20\%$ KV cache budget.

| Tasks | Methods | 5-shots | | 10-shots | |
|---|---|---|---|---|---|
| | | OPT-30B | OPT-66B | OPT-30B | OPT-66B |
| OpenBookQA | Full | 43.20 | 44.40 | 43.00 | 44.80 |
| | Local | 25.20 | 30.60 | 26.60 | 38.80 |
| | $H_2O$ | 43.00 | 44.20 | 42.80 | 44.80 |
| COPA | Full | 85.00 | 83.00 | 86.00 | 85.00 |
| | Local | 48.00 | 59.00 | 60.00 | 76.00 |
| | $H_2O$ | 84.00 | 82.00 | 85.00 | 86.00 |
| MathQA | Full | 26.23 | 27.87 | 26.67 | 27.00 |
| | Local | 20.87 | 25.49 | 21.11 | 23.08 |
| | $H_2O$ | 26.87 | 27.67 | 26.47 | 27.30 |

## C.8 Heavy-Hitter in Attention Blocks

The distribution of accumulated attention scores of all the tokens within attentions blocks is illustrated in Figure 10. We can observe that $H_2$ broadly exists in each layer.

## C.9 Comparsion with SpAtten

Compared with SpAtten [32], the main differences of $H_2O$ are i) They accumulate attention scores across attention heads and layers, while in our algorithm, each token can be kept or evicted independently across heads and layers, providing more flexibility for selecting critical KV embeddings; ii) We also use KV of the most recent tokens during generation and demonstrate that such $H_2$ tokens can effectively enhance other sparse attention strategies; iii) we formulate the KV cache eviction as a dynamic submodular problem and prove a theoretical guarantee (under mild assumptions) for our novel algorithms. Moreover, we also provide a quantitative comparison with SpAtten, and the results are reported in Table 11.

Table 11: Comparison between SpAtten [32] and $H_2O$ across various tasks with OPT-30B.

| Models | COPA | OpenBookQA | PiQA |
|---|---|---|---|
| Full | 85.00 | 43.20 | 78.51 |
| SpAtten | 82.00 | 41.90 | 77.06 |
| $H_2O$ | 84.00 | 43.00 | 78.45 |

## C.10 Heavy-Hitter in MLP Blocks

Besides the attention blocks, the presence of Heavy-Hitters ($H_2$) is observed within the MLP blocks of LLMs. We utilize the Wiki-Text-103 dataset as the input and record the activated frequency of neurons in the hidden layer of MLP blocks. As depicted in Figure 11, the activated frequency of neurons follows a power-law distribution, wherein a small number of neurons are activated by nearly all input tokens (with a $100\%$ frequency) while the majority of other neurons are rarely activated.

Subsequently, a thorough examination of various characteristics pertaining to $H_2$ in MLP blocks is conducted, encompassing the following aspects: (1) The elimination of $H_2$ leads to a substantial decline in performance, although such degradation can be easily recovered even with a mere $1\%$ of the training data; (2) $H_2$ exhibits a significant degree of overlap across different type of input content; (3) The emergence of $H_2$ occurs early in the training process, thus exhibiting an "early-bird" characteristic, and their positions undergo gradual changes during subsequent training phases.

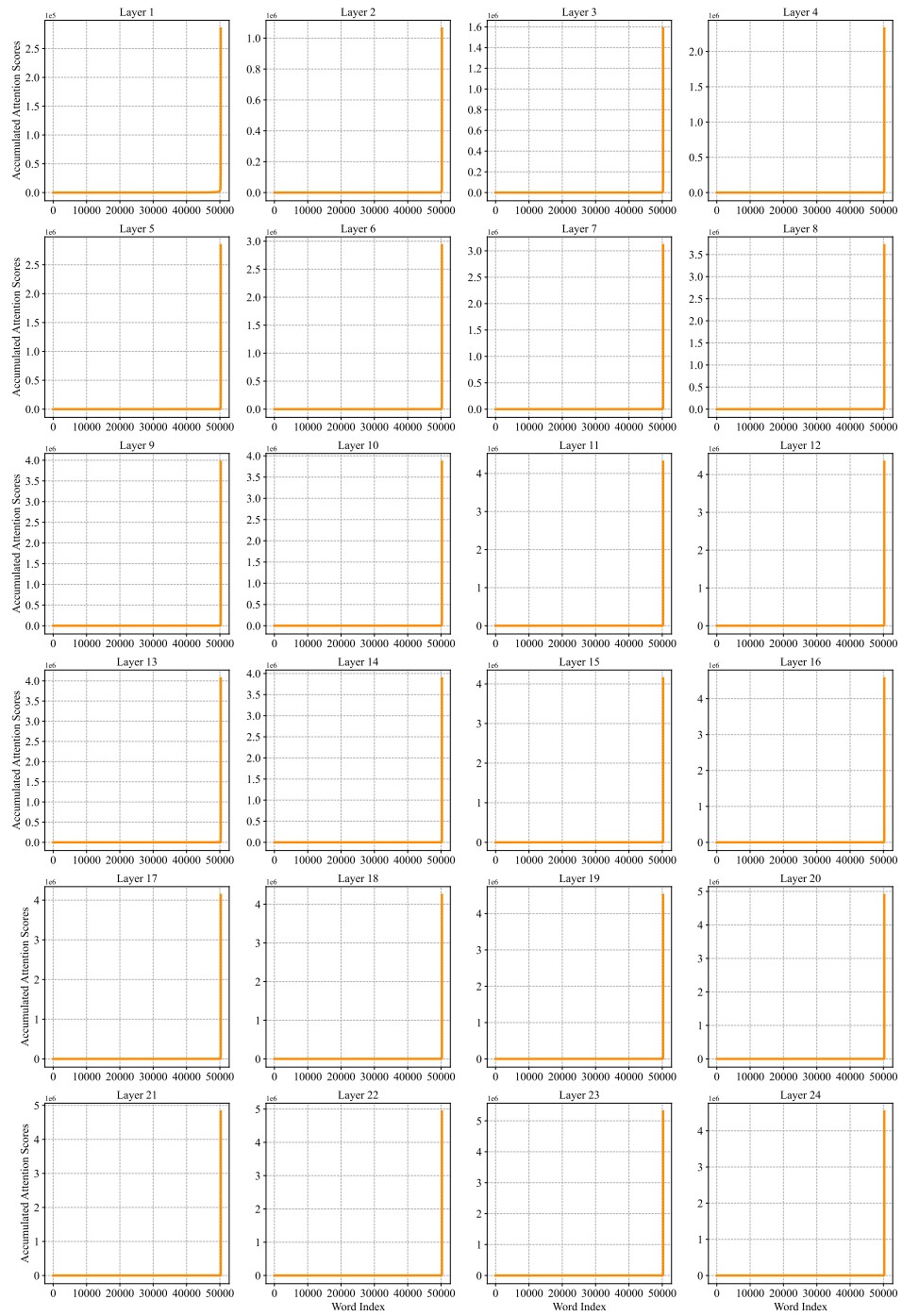

Figure 10: The distribution of accumulated attention scores with respect to the corresponding word. The x-axis represents the word index in the vocabulary, and the y-axis represents the accumulated attention score. Results are obtained from OPT-1.3B.

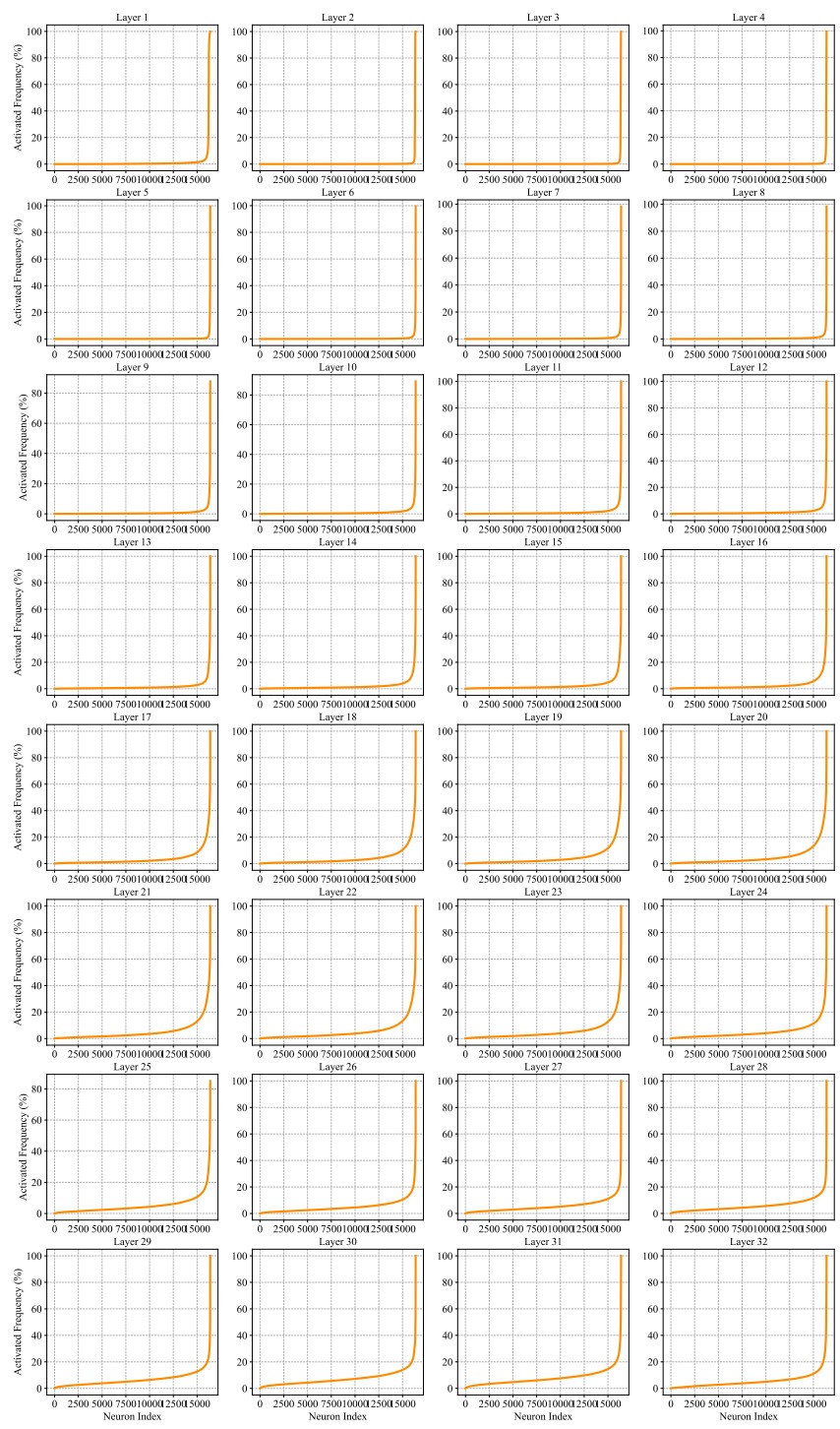

Figure 11: The emergence of $H_2$ in MLP blocks of OPT-6.7B. The x-axis represents the index of neurons in the hidden layers of MLP blocks, and the y-axis represents the activated frequency.

**Elimination of** $H_2$**.** We first train a GPT-2 using Wiki-Text-103 dataset and subsequently identify and prune the neurons exhibiting an activation frequency exceeding $20\%$ (*i.e.*, $H_2$). This pruning operation leads to a substantial decline in performance, as evidenced by an increase in perplexity from 19.32 to 31.78. The results emphasize the criticality of $H_2$ in preserving the functionality of the model. To assess the recoverability of the discarded information, we conduct a few-shot fine-tuning experiment, and the results are summarized in Table 12. The pruned model is fine-tuned with varying ratios of training data for 500 iterations, and it successfully regains performance levels equivalent to those of the pre-trained model. In contrast, when training the model from scratch using only $1\%$ of the training data, the resulting model achieves a perplexity of 554.12 only. These findings demonstrate that the knowledge encoded in $H_2$ can be easily restored.

Table 12: . Perplexity on the test-set of Wiki-Text-3 with GPT-2.

| Settings | 1% | 10% | 40% | 100% |
|---|---|---|---|---|
| Pretrained Model | | 19.32 | | |
| Remove $H_2$ | | 31.78 | | |
| Fine-tuning | 19.86 | 19.84 | 19.76 | 19.83 |

**Overlap across Diverse Input Contents.** Moreover, we conduct a comparative analysis of the activation frequencies acquired from various input contents. Specifically, utilizing the pretrained OPT-1.3B model, we evaluate three datasets, namely Wiki-Text-103, Penn Treebank, and Amazon Review. The positioning of $H_2$ is depicted in Figure 12, revealing significant concurrence across multiple datasets.

**Early-Bird Property.** Furthermore, our investigation reveals that $H_2$ displays an "early-bird" characteristic, as illustrated in Figure 13. By visualizing the distribution of activation frequencies across various checkpoints throughout the training process, we observe the emergence of a power-law behavior at an initial stage, specifically as early as a training budget of $4\%$. Subsequently, the positions of $H_2$ exhibit gradual and minimal changes.

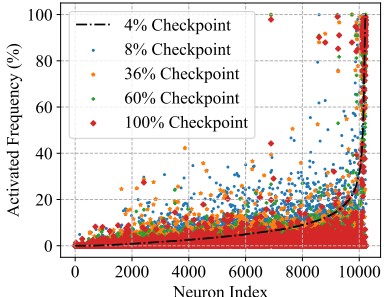

Figure 13: The distribution of activated frequency during training. Experiments are conducted with different checkpoints of OPT-2.7B during training.

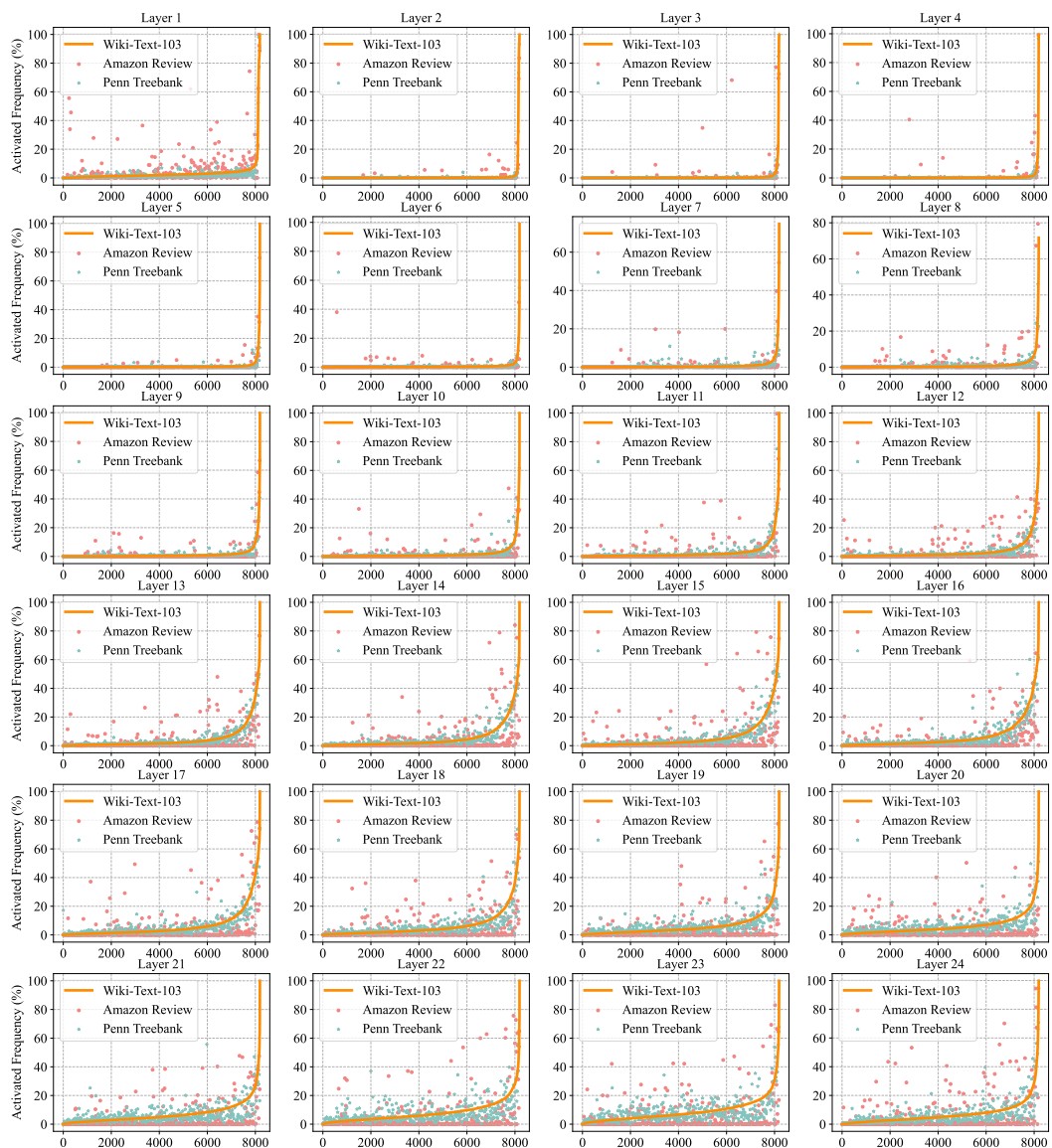

Figure 12: The distribution of activated frequency across diverse input content. The x-axis represents the index of neurons, which is ordered by the activated frequency from Wiki-Text-103.

# D Theoretical Analysis

Recently, a number of works have studied the attention scheme in LLMs from a theoretical perspective [96, 97, 98, 99, 100, 101, 102, 103, 104, 105, 106, 107, 108, 109, 110, 111, 112, 113, 114, 115, 116, 117, 118, 119, 120, 121, 122]. In this work, we provide a different and novel angle compared to the previous work. We present the concept of the submodular property and propose an eviction policy, known as greedy $H_2$, which is a modification of dynamic submodular maximization. Furthermore, assuming the attention scheme to be submodular, we establish that constructing the set $S_i$ without any cache size limitation satisfies the near-optimal property in terms of submodularity. We provide theoretical guarantees for our robust and approximate greedy eviction policy algorithm (Algorithm 2). Due to space limitation, we only give informal description of algorithm (Algorithm 2) in Section 4.1 In Section D.6, we give an algorithm (Algorithm 2) which has full and complete implementation details for Algorithm 1. We also offer a mathematical formulation for sparsity preservation that is observed in Section 3 and proposed an algorithm (Algorithm 4) to solve the problem.

Specifically, in Section D.1, we provide several basic definitions and notations. In Section D.2, we briefly the definition of submodular function. In Section D.3, we define the dynamic submodular framework, which gives the formal version of Definition 4.1. In Section D.4, we briefly review the static attention computation problem. In Section D.5, we formulate the attention computation in recursive fashion. In Section D.6, we briefly review our eviction policy, which gives the formal version of Definition 4.3. In Section D.7, we discuss the diminishing return for submodular. In Section D.8, we discuss the high-level ideas for submodular. In Section D.9, we analyze the robust greedy algorithm error propagation. In Section D.10, we explain how to add items into sets via approximate function. In Section D.11, we provide several definitions related to dynamic properties. In Section D.12, we prove an induction lemma for the exact function. In Section D.13, we prove an induction lemma for the approximation function. In Section D.14, we provide theoretical guarantees for both the full-knowledge version (formal version of Lemma 3.1) and the limited-cache-size version (formal version of Theorem 4.4). In Section D.15, we provide a more detailed discussion of theoretical work about attention computation and regression-related problems. In Section D.16, we provide a mathematical formulation for sparsity preserving. In Section D.17, we provide the definition of loss function which can potentially generate sparse (heavy hitter type attention sore). In Section D.18, we explain how to compute the gradient of the loss function. In Section D.19, we show how to compute the Hessian of the loss function. In Section D.20, we show that Hessian is positive definite. In Section D.21, we prove the Lipschitz property for the Hessian matrix. In Section D.22, we show that using a gradient-type algorithm is sufficient to optimize that (heavy hitter type) loss function.

## D.1 Notations

For a positive integer $n$, let $[n] := \{1, 2, \cdots, n\}$.

For a vector $x \in \mathbb{R}^n$, let $\sqrt{x} \in \mathbb{R}^n$ denote the vector with the $i$-th entry being $\sqrt{x_i}$ and $\mathrm{diag}(x) \in \mathbb{R}^{n \times n}$ denote the diagonal matrix with the $i$-th digonal entry being $x_i$. For two matrices $A, W \in \mathbb{R}^{n \times n}$, let $\|A\|_W := (\sum_{i=1}^n \sum_{j=1}^n W_{i,j} A_{i,j}^2)^{1/2}$ and $W \circ A$ denote the matrix where $(W \circ A)_{i,j} = W_{i,j} A_{i,j}$. For matrix $W \in \mathbb{R}^{n \times n}$, let $D_{W_i} := \mathrm{diag}(W_{i,:})$ with $i \in [n]$.

For two vectors $x \in \mathbb{R}^n$ and $w \in \mathbb{R}_{\geq 0}^n$, let $\|x\|_w := (\sum_{i=1}^n w_i x_i^2)^{1/2}$. For a vector $x$, its $\ell_2$ norm is defined as $\|x\|_2 := (\sum_{i=1}^n x_i^2)^{1/2}$ and its $\ell_p$ norm is defined as $\|x\|_p := (\sum_{i=1}^n |x_i|^p)^{1/p}$. For a square matrix $A$, we denote $\mathrm{tr}[A]$ as the trace of matrix $A$.

For a matrix $A \in \mathbb{R}^{n \times k}$ (suppose $n \geq k$), we use $\|A\|$ to denote its spectral norm, i.e., $\|A\| = \sup_x \|Ax\|_2 / \|x\|_2$. We use $\|A\|_F$ to denote its Frobenius norm $\|A\|_F := (\sum_{i=1}^n \sum_{j=1}^k A_{i,j}^2)^{1/2}$.

Suppose matrix $A \in \mathbb{R}^{n \times k}$ has SVD decomposition $U\Sigma V^\top$ where $U \in \mathbb{R}^{n \times k}$ (this matrix has orthonormal columns), $\Sigma \in \mathbb{R}^{k \times k}$ is a diagonal matrix, and $V \in \mathbb{R}^{k \times k}$. We call columns of $U$ singular vectors. We use $A^\dagger \in \mathbb{R}^{k \times n}$ to denote the Moore-Penrose pseudoinverse, then $A^\dagger = V\Sigma^{-1}U^\top$. Suppose $\Sigma \in \mathbb{R}^{k \times k}$ is sorted diagonal matrix, let $\sigma_1, \cdots, \sigma_k$ denote the diagonal entries of $\Sigma$. Then we call $\sigma_i$ the $i$-th singular value of the matrix, and we write it as $\sigma_i(A)$.

For any symmetric matrix $B \in \mathbb{R}^{k \times k}$, we denote its eigenvalue decomposition as $U \Lambda U^\top$, where $\Lambda$ is a diagonal matrix. Let $\lambda_1, \cdots, \lambda_k$ denote the entries on diagonal of $\Lambda \in \mathbb{R}^{k \times k}$. We say $\lambda_i$ is the $i$-th eigenvalue. Usually, we write it as $\lambda_i(B)$.

The connection between eigenvalues and singular values is

$$\sigma_i^2(A) = \lambda_i(A^\top A)$$

We use the notation $A \succeq 0$ to denote that matrix $A$ is positive semidefinite (psd). Mathematically, $A \succeq 0$ means for all vectors $x$, we have $x^\top A x \geq 0$.

Similarly, for two squarer matrices $A$ and $B$, we use $A \succeq B$ to denote the case where for all vectors $x$, $x^\top A x \geq x^\top B x$.

Let $\Pr[]$ and $\mathbb{E}[]$ denote the probability and expectation. We define the maximum between $a$ and $b$ as $\max\{a, b\}$. We denote $\min\{a, b\}$ (resp. $\max\{a, b\}$) as the minimum (reps. maximum) between $a$ and $b$.

Throughout, for non-negative real numbers a and b, we use the notation $a = (1 \pm \epsilon)b$ if $a \in [(1 - \epsilon)b, (1 + \epsilon)b]$.

### D.2 Submodular

We provide the standard definition of the submodular function.

**Definition D.1** (Submodular function [123]). *For a finite set $\Omega$, a submodular function is defined as $f : 2^\Omega \to \mathbb{R}$, where $2^\Omega$ denotes the power set of $\Omega$. It is characterized by the fulfillment of any of the following equivalent criteria:*

- *Condition 1.*

  - *For $S, T, \subseteq \Omega$ with $S \subseteq T$ and every $a \in \Omega \backslash T$ we have that $f(S \cup \{a\}) - f(S) \geq f(T \cup \{a\}) - f(T)$*

- *Condition 2.*

  - *For every $S, T \subseteq \Omega$ we have that $f(S) + f(T) \geq f(S \cup T) + f(S \cap T)$.*

- *Condition 3.*

  - *For every $S \subseteq \Omega$ and $a_1, a_2 \in \Omega \backslash S$ such that $a_1 \neq a_2$ we have that $f(S \cup \{a_1\}) + f(S \cup \{a_2\}) \geq f(S \cup \{a_1, a_2\}) + f(S)$.*

For convenience of discussion, in this paper, we always choose $\Omega = [n]$ when we want to discuss the submodular function.

Next, we provide some examples/types of submodular functions. One important class is called monotone,

**Definition D.2** (Monotone). *A set function $f$ is monotone if for every $T \subseteq S$ we have $f(T) \leq f(S)$.*

Here are a number of monotone submodular functions

- Linear (Modular) functions
  - A linear function can be represented as $f(S) = \sum_{i \in S} w_i$. If all the weights $w_i$ are nonnegative, then the function $f$ is considered monotone.
- Budget-additive functions
  - A budget-additive function has the form $f(S) = \min B, \sum_{i \in S} w_i$ where each weight $w_i$ and the budget $B$ are nonnegative.
- Coverage functions
  - We have a set $\Omega = \{E_1, E_2, \cdots, E_n\}$ where each $E_i$ is a subset of a broader set $\Omega'$. The coverage function can be expressed as $f(S) = | \cup_{E_i \in S} E_i|$ for any $S \subset \Omega$. This function can be generalized by assigning non-negative weights to the elements.

### D.3 Dynamic Submodular

The standard submodular function is mapping $2^{[n]}$ to real. Here we need a more general definition that maps $2^{[n]} \times 2^{[n]}$ to real.

**Definition D.3** (Strong submodular). *We define function $F : 2^{[n]} \times 2^{[n]} \to \mathbb{R}$, then for any set $Z \subset [n]$, we assume that $F(Z, \cdot) : 2^{[n]} \to \mathbb{R}$ is a submodular function.*

In fact, the above definition is stronger than we want, what we really need can be written as follows. We remark that in Definition 4.1 we provide an informal definition. Here we provide a more detailed version of the definition.

**Definition D.4** (Dynamic submodular framework, formal version of Definition 4.1). *Define function*

$$F : 2^{[n]} \times [n] \times 2^{[n]} \to \mathbb{R}.$$

*Then for any index $i \in [n]$, any set $Z \subseteq [i-1]$, we assume that*

$$F(Z, i, \cdot) : 2^{[n]} \to \mathbb{R}$$

*is a submodular function w.r.t. to $Z$, i.e.,*

- *For all sets $X, Y \subset [n]$ satisfy that $Z \subset X \subset Y$,*

- *For all element $x \in [n]$ satisfy that $x \in [n] \backslash Y$,*

*we have*

$$f_{Z,i}(X \cup \{x\}) - f_{Z,i}(X) \geq f_{Z,i}(Y \cup \{x\}) - f_{Z,i}(Y),$$

*where $f_{Z,i}(\cdot) := F(Z, i, \cdot)$.*

**Remark D.5.** *We remark that Definition D.4 is a weaker version of Definition D.3. We also like to mention that the informal definition (see Definition 4.1) only contains two input parameters, but in fact we need three input parameters (see Definition D.4).*

*In the later, when we use $f_{Z,i}(\cdot)$, we will replace $Z$ by $S_i$ for convenient of analysis, for example see Definition D.23, Definition D.24, Definition D.25 and Definition D.26.*

### D.4 Static Attention

Before we describe the recursive attention computation, we will first describe the static version of attention computation as follows (for examples, see Definition 1.1 in [98] and others [97, 100, 101, 106, 103, 118, 108]):

**Definition D.6.** *Given three matrices $Q, K, V \in \mathbb{R}^{d \times d}$, the goal is to compute*

$$\mathsf{Att}(Q, K, V) := D^{-1} A \cdot V$$

*where square matrix $A \in \mathbb{R}^{n \times n}$ can be rewritten as follows*

$$A = \exp(QK^\top)$$

*and diagonal matrix $D \in \mathbb{R}^{n \times n}$ can be written as follows*

$$D = \mathrm{diag}(A\mathbf{1}_n)$$

*Here we apply $\exp()$ to a matrix entry-wisely to a matrix. $\mathbf{1}_n$ is a length-$n$ vector where all the entries are ones. The operator $\mathrm{diag}()$ is turning a vector into a diagonal matrix.*

### D.5 Recursive Attention Definition

We first provide some definitions.

**Definition D.7.** *Given a query matrix $Q \in \mathbb{R}^{n \times d}$, we use $Q_{i,*}$ to denote a length-$d$ row vector that represents the $i$-th row of $Q$ for each $i \in [n]$.*

**Definition D.8.** *Given a key matrix $K \in \mathbb{R}^{n \times d}$, we use $K_{\leq i,*}$ to denote a $i \times d$ matrix that selects the first $i$ rows from $K$.*

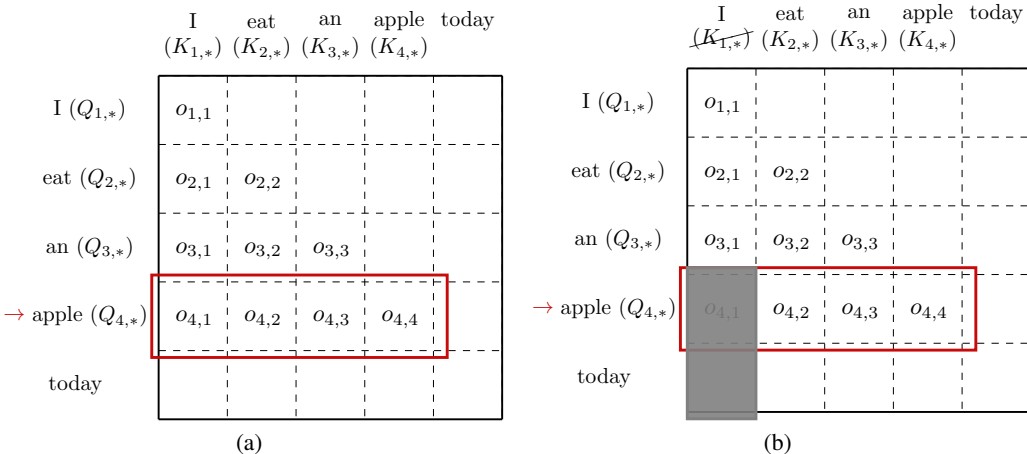

Figure 14: (a) Exact version of the attention computation. Here, our example is about the language modeling task. At this stage, the model predicts the word 'apple' and computes the exact attention vector $o_4$. (b) Approximate version of the attention computation. Let the budget of the number of tokens we can track be 3. We truncate the key matrix for the first token $K_{1,*}$ and only keep $K_{2,*}$, $K_{3,*}$ and $K_{4,*}$ in the memory. Compared with the exact version, we don't compute $o_{4,1}$.

Next, we show the exact version of computing attention.

**Definition D.9** (Exact Version). *Given $Q, K \in \mathbb{R}^{n \times d}$*

- *For each $i \in [n]$, we use $o_i \in \mathbb{R}^i$ to denote a length-$i$ vector.*
  - *For each $j \in [i]$, we use $o_{i,j}$ to denote the $j$-th coordinate of vector $o_i \in \mathbb{R}^i$*

*For the first layer, we have*

- $o_{1,1} = 1$.

*For the second layer, we have*

- *length-2 vector* $o_2 := \underbrace{D_2^{-1}}_{\text{scalar}} \cdot \exp\big(\underbrace{Q_{2,*}}_{1 \times d} \underbrace{(K_{\leq 2,*})^\top}_{d \times 2}\big)$

- *scalar* $D_2 := \underbrace{\exp(Q_{2,*}(K_{\leq 2,*})^\top)}_{1 \times 2} \cdot \underbrace{\mathbf{1}_2}_{2 \times 1}$

*For each $i \in [n]$, for the $i$-th layer, we have*

- *length-$i$ vector* $o_i := \underbrace{D_i^{-1}}_{\text{scalar}} \cdot \exp\big(\underbrace{Q_{i,*}}_{1 \times d} \underbrace{(K_{\leq i,*})^\top}_{d \times i}\big)$

- *scalar* $D_i := \underbrace{\exp(Q_{i,*}(K_{\leq i,*})^\top)}_{1 \times i} \cdot \underbrace{\mathbf{1}_i}_{i \times 1}$

Now, we show the approximate version of computing attention. Instead of computing the entire attention $o_i$, we only compute the attention of the tokens that are being tracked.

**Definition D.10** (Approximate Version). *Given $Q, K \in \mathbb{R}^{n \times d}$*

- *For each $i \in [n]$, we use $o_i \in \mathbb{R}^i$ to denote a length-$i$ vector.*
  - *For each $j \in [i]$, we use $o_{i,j}$ to denote the $j$-th coordinate of vector $o_i \in \mathbb{R}^i$*

- *Let $k \in [n]$ be the budget of the number of tokens we can track (due to the memory issue).*

*For the first layer, we have*

- $o_{1,1} = 1$.

*For the second layer, we have*

- *length-2 vector* $o_2 := \underbrace{D_2^{-1}}_{\text{scalar}} \cdot \exp(\underbrace{Q_{2,*}}_{1 \times d} \underbrace{(K_{\leq 2,*})^\top}_{d \times 2})$

- *scalar* $D_2 := \underbrace{\exp(Q_{2,*}(K_{\leq 2,*})^\top)}_{1 \times 2} \cdot \underbrace{\mathbf{1}_2}_{2 \times 1}$

*For each $i \in [n]$, for the $i$-th token, we have*

- *Let $S_i \subset [n]$ denote the tokens we're tracking when we are predicting the $i$-th token.*

- $|S_i| = k$

- $|S_i \backslash S_{i-1}| \leq 1$ *or equivalently* $|S_i \cap S_{i-1}| \geq k - 1$

- *length-$i$ vector* $o_i := \underbrace{D_i^{-1}}_{\text{scalar}} \cdot \exp(\underbrace{Q_{i,*}}_{1 \times d} \underbrace{(K_{S_i,*})^\top}_{d \times i})$

- *scalar* $D_i := \underbrace{(\exp(Q_{i,*}(K_{S_i,*})^\top) - 1_{[i] \backslash S_i})}_{1 \times i} \cdot \underbrace{\mathbf{1}_i}_{i \times 1}$

In a certain sense, the above definition is related to finding heavy hitters in compressive sensing (for more background on compressive sensing, we refer readers to [124, 125, 126]).

## D.6 Eviction Policy

The goal of this section is to our eviction policy under space limitation. We start by giving a process without having space limitations. We denote the attention query matrix as $Q \in \mathbb{R}^{n \times d}$ and the key matrix as $K \in \mathbb{R}^{n \times d}$. $Q_{i,*}$ represents the $i$-th row of $Q$ and $K_{\leq i,*}$ represents the first $i$ rows of $K$. For simplicity, $K_{S_i,*}$ denotes a sub-matrix of $K$ which selects $\bar{S}_i$ rows from $K$.

**Definition D.11** (The generative process with Full knowledge). *For each $i \in [n]$, for the $i$-th token, we have*

- *length-$i$ vector* $o_i := D_i^{-1} \cdot \exp(Q_{i,*}(K_{\leq i,*})^\top)$, *which denotes the attention of the $i$-th word.*

- *scalar* $D_i := \exp(Q_{i,*}(K_{\leq i,*})^\top) \cdot \mathbf{1}_i$, *which denotes the normalization factor.*

In the above process, since we have no space limitation, then we can keep all the scores. In the following process, we show how to handle things when there is a space limitation. In this case, we need to dynamically maintain set $S_i$ such that $|S_i| \leq k$ (Compared to the full knowledge case, we can think of $|S_i| = i$).

**Definition D.12** ($H_2O$ Eviction Policy, Formal version of Definition 4.3). *Let $k$ denote the budget of space and $k < n$. Let $F_{\text{score}} : 2^{[n]} \rightarrow \mathbb{R}$ denote certain score function. Let $S_{i-1}$ denote the source set. Let $S_i$ denote the target set. We defined the eviction policy $g : S_{i-1} \rightarrow S_i$ s.t.*

- $|S_i| \leq k$ *(KV cache size is not changing over the time)*

- $|S_i \backslash S_{i-1}| \leq 1$ *(we can evict at most 1 KV in the KV cache)*

- *We construct $S_i \leftarrow (S_{i-1} \cup \{i\}) \backslash \{u\}$ as $u \leftarrow \arg\max_{v \in (S_{i-1} \cup \{i\})} F_{\text{score}}(S_{i-1} \cup \{i\} \backslash \{v\})$*

**Remark D.13.** *We remake that, in Definition 4.3, we introduce a simplified notation where we state that $|S_i| = k$ in the first bullet point, but in general, we consider the case where $|S_i| \leq k$ for the first $k$ tokens. To accommodate this more general scenario, we present Definition D.12, which provides a broader definition that handles the situation where $|S_i| \leq k$.*

**Algorithm 2** H$_2$ Eviction Algorithm, Query matrix $Q \in \mathbb{R}^{n \times d}$, key matrix $K \in \mathbb{R}^{n \times d}$, budget size of KV cache $k \in \mathbb{N}$. Formal and detailed version of Algorithm 1.

---

1: **procedure** H$_2$_EVICTION($Q \in \mathbb{R}^{n \times d}, K \in \mathbb{R}^{n \times d}, k \in \mathbb{N}$)
2:     $S_0 \leftarrow \emptyset, \widetilde{o}_0 \leftarrow 0$                                            ▷ Initialization the algorithm
3:     **for** $i = 1 \rightarrow n$ **do**                                            ▷ Linear scan each token
4:         **if** $i \leq k$ **then**                                      ▷ If the KV cache is not full
5:             $S_i \leftarrow S_{i-1} \cup \{i\}$                               ▷ Expand the set directly
6:         **else**                                                ▷ If the KV cache is full
7:             $D_i \leftarrow (\exp(Q_{i,*}(K_{S_{i-1},*})^\top) - 1_{[i] \setminus S_{i-1}}) \cdot \mathbf{1}_i$   ▷ Compute the normalization factor
8:             $o_i \leftarrow D_i^{-1} \cdot \exp(Q_{i,*}(K_{S_{i-1},*})^\top)$                ▷ Compute score vector
9:             $\widetilde{o}_i \leftarrow \widetilde{o}_{i-1} + o_i$                  ▷ Accumulate the scores (Remark D.15)
10:            Let score function be $F_{\text{score}}(T) := h(\sum_{s \in T} \widetilde{o}_{i,s})$ ▷ Score function (Remark D.14)
11:            $u \leftarrow \arg\max_{v \in (S_{i-1} \cup \{i\})} F_{\text{score}}(S_{i-1} \cup \{i\} \setminus \{v\})$     ▷ Find an entry to swap
12:            $S_i \leftarrow (S_{i-1} \cup \{i\}) \setminus \{u\}$                     ▷ Construct $S_i$
13:         **end if**
14:     **end for**
15: **end procedure**

---

**Remark D.14.** *We remark the above function $F_{\text{score}}$ can have multiple choices. In the later analysis, we provide two choices. If we use the exact function, then $F_{\text{score}}$ is $f_{S_i,i}$ (see Definition D.23 and Definition D.25). If we use the approximate function, then $F_{\text{score}}$ if $\widetilde{f}_{S_i,i}$ (see Definition D.24 and Definition D.26). Let $h : \mathbb{R} \to \mathbb{R}$ (the $h$ being used Line 10 in Algorithm 2) denote function. We hope to choose $h$ such that it gives the submodular property for the function $F_{\text{score}}$ in the sense of our dynamic framework. For example, $h(z)$ is usually chosen to be some non-decreasing concave function such as $\sqrt{z+1}$ and $\log(z+1)$. For simplicity, in Algorithm 1 (which is the informal version of Algorithm 2), we choose $h(z) = z$ for the purpose of providing readers with an intuitive understanding and due to space limitation.*

**Remark D.15.** *We remark that, in Line 10 in Algorithm 2, the $\widetilde{o}_{i,s}$ is the accumulation of attention score for the token $s$ in set $T$. In our Algorithm 1, we only use $o_s$ in Line 10 (see Algorithm 1) to represent that accumulation of the attention score for simplicity.*

*In Algorithm 2, for simplicity of the presentation, we can create multiple vectors $\widetilde{o}_0, \cdots \widetilde{o}_n$ (see Line 2, Line 9, Line 10). However, every time at $i$-th, we only need to use the information for $\widetilde{o}_i$ which has at most length $n$ size. Thus, a straightforward and better implementation for only using one length-$n$ to store accumulative score can reduce $n^2$ usage to $n$.*

### D.7 Explaining Submodular Diminishing Return Property in Attention Scheme

In the standard submodular problem [123], we are given a ground set $[n]$ and a function $f : 2^{[n]} \to \mathbb{R}$. The goal is to find a set of elements $S$ such that $f(S)$ is maximized.

We say function $f$ is submodular (Recall the formal definition in Definition D.1), if for every $X, Y \subset [n]$ with $X \subseteq Y$ and every $x \in [n] \setminus Y$ we have

$$f(X \cup \{x\}) - f(X) \geq f(Y \cup \{x\}) - f(Y)$$

Submodular functions can represent the cost of items, due to their diminishing returns property [127, 128]. It suggests that the increase in information obtained from selecting a candidate object, such as a word or sentence, becomes smaller as more objects have already been chosen for the summary.

For instance, we introduce a new token, denoted as "w", into two sets, $S$ and $S_0$, where the concepts covered by $S_0$ are a subset of those covered by $S$. By intuition, the information added to $S_0$ by "w" should be larger compared to adding it to $S$, as the new concepts carried by "w" might have already been covered by the concepts present in $S$ but not in $S_0$. This property is known as the diminishing return property. Hence, we propose that the neural coverage function [129] should exhibit a desirable characteristic called submodularity.

## D.8 Submodular: High Level Ideas

We formalize the submodular function maximization problem with cardinality constraint in this section. Informally, a set function is submodular if it has decreasing marginal increment.

**Definition D.16** (Submodular function). *We denote $f : 2^{[n]} \to \mathbb{R}$ as a set function. The discrete derivative $\Delta_f$ is defined as follows:*

$$\Delta_f(i \mid S) := f(S \cup \{i\}) - f(S).$$

*A function $f$ is submodular if, for any $S \subseteq T$ and $i \in [n] \backslash T$, the following inequality holds:*

$$\Delta_f(i \mid T) \leq \Delta_f(i \mid S).$$

For simplicity, we present the problem of maximizing a submodular function with a cardinality constraint (1). Our goal is to solve the optimization problem efficiently.

$$\begin{aligned} \max_{S \subseteq [n]} \quad & f(S) \\ \text{s.t.} \quad & |S| \leq k \end{aligned} \tag{1}$$

**Representation of $f(S)$**   One challenge in designing the algorithm is determining the representation of input instances. Since the constraint in optimization problem (1) is straightforward, we need to decide how to represent $f(S)$. Suppose $S = i_1, i_2, \cdots, i_m \subseteq [n]$, we can decompose $f(S)$ into a sum of increments as follows:

$$f(S) = f(S_0) + \sum_{j=1}^{m} (f(S_j) - f(S_{j-1})), \tag{2}$$

where $S_0 = \emptyset$ and $S_j = S_{j-1} + i_j$. Without loss of generality, we assume $f(\emptyset) = 0$. By the definition of $\Delta_f(i|S)$, we have $f(S_j) - f(S_{j-1}) = \Delta_f(i_j|S_{j-1})$. Therefore, the decomposition (2) can be simplified as follows:

$$f(S) = \sum_{j=1}^{m} \Delta_f(i_j|S_{j-1}) \tag{3}$$

To introduce our advanced data structure later, we further represent $\Delta_f(i|S)$ in the form of

$$\Delta_f(i|S) = u_i^\top h(S) u_i \tag{4}$$

where $u_i \in \mathbb{R}^d$ is a $d$-dimensional vector and $h(S) \in \mathbb{R}^{d \times d}$ is a $d$-by-$d$ matrix.

In practice, a significant subclass of submodular functions is the monotone submodular functions, i.e. functions $f$ satisfying $f(A) \leq f(B)$ for all $A \subseteq B \subseteq [n]$. When $f$ is monotone, we could restrict all $h(S)$ to be positive semidefinite (PSD) matrixes. When the matrix $h(S)$ is positive semidefinite (PSD), it makes us achieve a faster acceleration.

## D.9 Robust Greedy with Error Propagation

The procedure for the greedy selection algorithm initiates with empty set $S_0 = \emptyset$. For each iteration in the main loop, the algorithm chooses the element that maximizes the marginal increment to add to the set. When the size of the set eventually reaches $k$, the algorithm return that set $S$. Specifically, for iteration $t \in \{1, 2, \cdots, k\}$, we let

$$S_t \leftarrow S_{t-1} \cup \{j_t\}$$

where the element in the singleton is $j_t = \arg\max_{j \in \overline{S}_{t-1}} f(S_{t-1} \cup \{j\})$. The greedy strategy is effective in the sense that the approximation error of it is $1 - 1/e$.

**Theorem D.17** ([130]). *For a monotone submodular function $f$, the greedy algorithm (Algorithm 3) guarantees to output a set $S$ satisfying*

$$f(S) \geq (1 - 1/e) \max_{|T|=k} \{f(T)\}.$$

**Corollary D.18** ([131]). *Given*

---
**Algorithm 3** Greedy algorithm benchmark
---
1: **procedure** GREEDYALGORITHM(submodular function $f$)
2:    $S_0 \leftarrow \emptyset$                                                   $\triangleright$ Initialize an empty set $S_0$
3:    **for** $t = 0 \rightarrow k - 1$ **do**
4:        $j \leftarrow \arg\max_i \{f(S_t \cup \{i\})\}$    $\triangleright$ Find the element $i \in [n]$ that maximize $f(S_t \cup \{i\})$
5:        $S_{t+1} \leftarrow S_t \cup \{j\}$                $\triangleright$ Append element $i$ to set $S_t$ to get $S_{t+1}$
6:    **end for**
7:    **return** $S_k$
8: **end procedure**
---

- *accuracy parameter $\epsilon > 0$.*

- *integer $k \geq 1$*

- *Let $O$ denote an oracle that takes an arbitrary set $S \subseteq [n]$ and $i \in [n]\backslash S$, returns a value $O(S, i)$ with guarantee that $\Delta(i|S) - \epsilon \leq O(S, i) \leq \Delta(i|S) + \epsilon$.*

- *Let $A$ denote an algorithm that each time step $t = 1, 2, \cdots, k$, it selects $j_t = \arg\max_j \{O(S_{t-1}, j)\}$ and lets $S_t \leftarrow S_{t-1} \cup \{j_t\}$.*

*Then this algorithm $A$*

- *returns a set $S_k$*

- *the $S_k$ satisfy that*

$$f(S_k) \geq (1 - 1/e) \max_{|T|=k} \{f(T)\} - k(2 - 1/e)\epsilon.$$

## D.10 Robust Submodular and Adding Items

We first propose a lemma that shows the robustness of the submodular.

**Lemma D.19.** *Suppose that for each $i \in [n]$, we have*

- *for all $S_i \subset [n]$ with $|S_i| \leq k$*

- *for all $X \subset [n]$, $X \subseteq S_i \cup \{i\}$ and $|X \cap S_i| \leq |S_i|$,*

$$|(\widetilde{f}_{S_i,i}(X)) - \widetilde{f}_{S_i,i}(S_i) - (f_{S_i,i}(X) - f_{S_i,i}(S_i))| \leq \epsilon/(2n)$$

*Let $\mathsf{opt}_i$ denote the optimal cost that can be achieved by using $f$ in $i$-th iteration. If the greedy algorithm use $f$ to find the solution has performance at least $(1 - 1/e) \cdot \mathsf{opt}_i$, then using $\widetilde{f}$, we can obtain a solution that has performance at least $(1 - 1/e) \cdot \mathsf{opt}_i - \epsilon$*

*Proof.* The proof follows from Lemma D.18. $\qquad\square$

Next, we explain how to add items into sets based on exact values.

**Definition D.20** (Expanding items based on exact value)**.** *If the following conditions hold*

- *Let $S_i \subseteq [i - 1]$.*

- *Let $f_{S_i,i} : 2^{[i]} \rightarrow \mathbb{R}$.*

*Then, we can define $S_{i+1}$ as follows*

- *If $|S_i| = k$ then $S_{i+1} = S_i \cup \{i\}\backslash u$ where $u = \arg\max_{v \in S_i \cup \{i\}} f_{S_i,i}(S_i \cup \{i\}\backslash v) - f_{S_i,i}(S_i)$*

- *If $|S_i| < k$, then $S_{i+1} = S_i \cup \{i\}$.*

**Remark D.21.** *We remark that $u = \arg\max_{v \in S_i \cup \{i\}} f_{S_i,i}(S_i \cup \{i\} \backslash v) - f_{S_i,i}(S_i)$ is the same as $u = \arg\max_{v \in S_i \cup \{i\}} f_{S_i,i}(S_i \cup \{i\} \backslash v)$. For the convenience of discussion and analysis, we will switch to using both cases in different places.*

Here, we explain how to add items into sets via approximate values.

**Definition D.22** (Expanding items based on approximate value). *If the following conditions hold*

- *Let $S_i \subseteq [i-1]$.*

- *Let $\widetilde{f}_{S_i,i} : 2^{[i]} \to \mathbb{R}$.*

*Then, we can define $S_{i+1}$ as follows*

- *If $|S_i| = k$ then $S_{i+1} = S_i \cup \{i\} \backslash u$ where $u = \arg\max_{v \in S_i \cup \{i\}} \widetilde{f}_{S_i,i}(S_i \cup \{i\} \backslash v) - \widetilde{f}_{S_i,i}(S_i)$*

- *If $|S_i| < k$, then $S_{i+1} = S_i \cup \{i\}$.*

## D.11 Universal Conditions

We state several definitions for both the exact function and approximation function.

**Definition D.23** (Universal Monotone Condition (for exact function)). *We say $f$ has universal monotone condition, if for all $i \in [n]$ for all $S_i \subseteq [i-1]$, we have*

$$f_{S_i,i}(X) \geq f_{S_i,i}(Y), \quad \forall Y \subset X$$

**Definition D.24** (Universal Monotone Condition (for approximate function)). *We say $\widetilde{f}$ has universal monotone condition, if for all $i \in [n]$ for all $S_i \subseteq [i-1]$, we have*

$$\widetilde{f}_{S_i,i}(X) \geq \widetilde{f}_{S_i,i}(Y), \quad \forall Y \subset X$$

**Definition D.25** (Universal Dynamic Condition 1(for exact function)). *We say $f$ has universal dynamic condition 1(for exact function), if for all $i \in [n]$ for all $S_i \subseteq [i-1]$, $S_{i-1} \subseteq [i-2]$, $|S_i \backslash S_{i-1}| \leq 1$, we have*

$$f_{S_i,i}(S_i) \geq (1-\theta) \cdot f_{S_{i-1},i-1}(S_i)$$

**Definition D.26** (Universal Dynamic Condition 1 (for approximate function)). *We say $\widetilde{f}$ has universal dynamic condition 1(for approximate function), if for all $i \in [n]$ for all $S_i \subseteq [i-1]$, $S_{i-1} \subseteq [i-2]$, $|S_i \backslash S_{i-1}| \leq 1$, we have*

$$\widetilde{f}_{S_i,i}(S_i) \geq (1-\theta) \cdot \widetilde{f}_{S_{i-1},i-1}(S_i)$$

We define $\mathrm{opt}_i$ as follows:

**Definition D.27.** *Let $k$ denote the budget length. For each $i \in [n]$, we define $\mathrm{opt}_i$ as*

$$\max\{f_{X,i}(Y) \mid X \subseteq [i-1], Y \subseteq [i], |X| \leq k, |Y| \leq k, |Y \backslash X| \leq 1\}.$$

**Definition D.28** (Universal Dynamic Condition 2). *For $i \in [n]$, we define $\mathrm{opt}_i$ as Definition D.27. We say it has universal dynamic condition if for each $i \in [n]$, we have*

$$\mathrm{opt}_i \geq (1-\gamma) \cdot \mathrm{opt}_{i-1}.$$

## D.12 Induction Lemma for Exact Function

The goal of this section is to prove Lemma D.29

**Lemma D.29** (Induction Lemma). *For a fixed $i$, suppose the following conditions hold*

- **Set Condition.** $S_i \subseteq [i-1]$

- **Budget Condition.** $|S_i| \leq k$

- **Value Condition.** $f_{S_{i-1},i-1}(S_i) \geq (1 - 1/e) \cdot (1 - \theta)^i (1 - \gamma)^i \mathsf{opt}_i$

- **Universal Dynamic Condition 1 (for exact function).** *(See Definition D.25)* $f_{S_i,i}(S_i) \geq (1 - \theta) \cdot f_{S_{i-1},i-1}(S_i)$

- **Universal Dynamic Condition 2.** *(See Definition D.28)* $\mathsf{opt}_i \geq (1 - \gamma) \cdot \mathsf{opt}_{i+1}$

- **Universal Monotone Condition (for exact function).** *(See Definition D.23)* $f_{S_i,i}(X) \geq f_{S_i,i}(Y)$ *for all* $Y \subset X$

*Then if we construct $S_{i+1}$ as Definition D.20, then we have*

- **Set Condition.** $S_{i+1} \subseteq [i]$

- **Budget Condition.** $|S_{i+1}| \leq k$

- **Value Condition.** $f_{S_i,i}(S_{i+1}) \geq (1 - 1/e) \cdot (1 - \theta)^{i+1} (1 - \gamma)^{i+1} \mathsf{opt}_{i+1}$

*Proof.* **Proof of Set Condition.**

Note that $S_i \subseteq [i - 1]$, by using the way we construct $S_{i+1}$, then it is obvious that $S_{i+1} \subseteq [i]$.

**Proof of Budget Condition.**

Note that $|S_i| \leq k$, by using the way we construct $S_{i+1}$, then it is straightforward that $|S_{i+1}| \leq k$.

**Proof of Value Condition.**

We can show that

$$
\begin{aligned}
f_{S_i,i}(S_{i+1}) &\geq f_{S_i,i}(S_i) \\
&\geq (1 - \theta) \cdot f_{S_{i-1},i-1}(S_i) \\
&\geq (1 - \theta) \cdot ((1 - 1/e)(1 - \theta)^i (1 - \gamma)^i \mathsf{opt}_i - i \cdot \epsilon_0) \\
&\geq (1 - \theta) \cdot ((1 - 1/e)(1 - \theta)^i (1 - \gamma)^{i+1} \mathsf{opt}_{i+1}) \\
&\geq (1 - 1/e) \cdot (1 - \theta)^{i+1} (1 - \gamma)^{i+1} \mathsf{opt}_{i+1}.
\end{aligned}
$$

where the first step follows from **Universal Monotone Condition**, the second step follows from **Universal Dynamic Condition 1**, the third step follows from **Value Condition**, the forth step follows from **Universal Dynamic Condition 2**, and the last step follows from simple algebra.

Thus, we complete the proof. $\square$

### D.13   Induction Lemma for Approximate Function

The goal of this section is to prove Lemma D.30

**Lemma D.30** (Induction Lemma). *For a fixed $i$, suppose the following conditions hold*

- **Set Condition.** $S_i \subseteq [i - 1]$

- **Budget Condition.** $|S_i| \leq k$

- **Value Condition.** $f_{S_{i-1},i-1}(S_i) \geq (1 - 1/e) \cdot (1 - \theta)^i (1 - \gamma)^i \mathsf{opt}_i - i \cdot \epsilon_0$

- **Universal Dynamic Condition 1 (for approximate function).** *(see Definition D.26)* $\widetilde{f}_{S_i,i}(S_i) \geq (1 - \theta) \cdot \widetilde{f}_{S_{i-1},i-1}(S_i)$

- **Universal Dynamic Condition 2.** $\mathsf{opt}_i \geq (1 - \gamma) \cdot \mathsf{opt}_{i+1}$

- **Universal Approximate Condition.** *(See Definition D.28)* $f_{S_i,i}(X) \geq \widetilde{f}_{S_i,i}(X) - \epsilon_0$ *for all $X$*

- **Universal Monotone Condition (for approximate function).** *(see Definition D.24)* $\widetilde{f}_{S_i,i}(X) \geq \widetilde{f}_{S_i,i}(Y)$ *for all $Y \subset X$*

*Then if we construct $S_{i+1}$ as Definition D.22, then we have*

- **Set Condition.** $S_{i+1} \subseteq [i]$

- **Budget Condition.** $|S_{i+1}| \leq k$

- **Value Condition.** $f_{S_i,i}(S_{i+1}) \geq (1 - 1/e) \cdot (1-\theta)^{i+1}(1-\gamma)^{i+1}\mathsf{opt}_{i+1} - (i+1) \cdot \epsilon_0$

*Proof.* **Proof of Set Condition.**

Note that $S_i \subseteq [i-1]$, by using the way we construct $S_{i+1}$, then it is obvious that $S_{i+1} \subseteq [i]$.

**Proof of Budget Condition.**

Note that $|S_i| \leq k$, by using the way we construct $S_{i+1}$, then it is straightforward that $|S_{i+1}| \leq k$.

**Proof of Value Condition.**

We can show that

$$
\begin{aligned}
f_{S_i,i}(S_{i+1}) &\geq \widetilde{f}_{S_i,i}(S_{i+1}) - \epsilon_0 \\
&\geq \widetilde{f}_{S_i,i}(S_i) - \epsilon_0 \\
&\geq (1-\theta) \cdot \widetilde{f}_{S_{i-1},i-1}(S_i) - \epsilon_0 \\
&\geq (1-\theta) \cdot ((1-1/e)(1-\theta)^i(1-\gamma)^i\mathsf{opt}_i - i \cdot \epsilon_0) - \epsilon_0 \\
&\geq (1-\theta) \cdot ((1-1/e)(1-\theta)^i(1-\gamma)^{i+1}\mathsf{opt}_{i+1} - i \cdot \epsilon_0) - \epsilon_0 \\
&\geq (1-1/e) \cdot (1-\theta)^{i+1}(1-\gamma)^{i+1}\mathsf{opt}_{i+1} - (i+1) \cdot \epsilon_0.
\end{aligned}
$$

where the first step follows from **Universal Approximate Condition**, the second step follows from **Universal Monotone Condition**, the third step follows from **Universal Dynamic Condition 1**, the forth step follows from **Value Condition**, the fifth step follows from **Universal Dynamic Condition 2**, and the last step follows from simple algebra.

Thus, we complete the proof. $\square$

### D.14 Theoretical Result

We first give the guarantee of our full-knowledge version (without cache size limitation).

**Lemma D.31** (Formal version of Lemma 3.1). *Under the mild assumption, let $k$ denote any target size. If we greedily compute the attention score based on full information, then we can find the set $S_i$ such that*

$$f(S_i) \geq (1 - 1/e) \cdot (1-\alpha)\mathsf{opt}_i,$$

*where $\alpha \in (0,1)$ are parameters.*

*Proof.* The proof follows from using Theorem D.17, Corollary D.18, Lemma D.29 with choosing $\theta = \gamma = \alpha/(10n)$. $\square$

Next, we show the guarantee for our robust and approximate greedy eviction policy algorithm (Algorithm 2).

**Theorem D.32** (Formal version of Theorem 4.4). *Under the mild assumption, let $k$ denote the budget of space limitation. If for each token, we greedily compute the attention score based on top-$k$ choice, then we can show the set $\widetilde{S}_i$ we generate each for token $i \in [n]$ satisfy that*

$$f(\widetilde{S}_i) \geq (1 - 1/e) \cdot (1-\alpha)\mathsf{opt}_i - \beta,$$

*where $\alpha \in (0,1), \beta > 0$ are parameters.*

*Proof.* The proof follows from using Theorem D.17, Corollary D.18, Lemma D.30 with choosing $\epsilon_0 = \beta/(10n)$ and $\theta = \gamma = \alpha/(10n)$. $\square$

### D.15 Extended Related Work for Theoretical Attention Problems

The static attention computation is asking the following question that given $Q, K, V \in \mathbb{R}^{n \times d}$, the goal is to $D^{-1} \exp(QK^\top) V$ where $D = \operatorname{diag}(\exp(QK^\top) \mathbf{1}_n)$. [98] studied the static attention computation from both algorithm and hardness. On the positive, they provide an almost linear time algorithm to approximately compute the attention matrix. On the negative side, assuming a strong exponential time hypothesis (SETH), they prove a hardness result. Their hardness result is, unless SETH fails, there is no algorithm that runs in truly subquadratic time to approximately compute the attention matrix. Further, [100] considers the dynamic of attention computation problem. They also provide both algorithmic results and hardness results. In the work of [103], they consider the sparsification of attention matrix construction. In particular, they assume that situation that $d \gg n$, and show how to sparsify the columns of matrix $Q$. [103] provides two algorithms, one is a randomized algorithm, and the other is a deterministic algorithm. Differential privacy is a famous and textbook topic in graduate school, recently the work of [101] shows how to give a differentially private algorithm for computing the attention matrix. For a given $A \in \mathbb{R}^{n \times d}$ and vector $b \in \mathbb{R}^n$, [104] formulates and studies exponential regression $\min_x \| \exp(Ax) - b \|_2$. Then [105] considers the normalization factor in exponential regression and defines the softmax regression problem $\min_x \| \langle \exp(Ax), \mathbf{1}_n \rangle^{-1} \exp(Ax) - b \|_2$. [107] moves the scaling factor from $\exp(Ax)$ to $b$ and defines a rescaled softmax regression problem $\min_x \| \exp(Ax) - \langle \exp(Ax), \mathbf{1}_n \rangle \cdot b \|_2$.

### D.16 Sparsity Preserving

Recall that in Figure 2, we observe that even when trained densely, the attention matrices of LLMs are over 95% sparse at inference time. Only 5% of the KV cache is sufficient for decoding the same output token at each generation step. Here, we provide some formal formulations for sparsity.

**Definition D.33.** *Suppose the following conditions*

- *Let $S_0 \subset [m]$.*
- *Let $k = |S_0|$.*
- *Let $\tau \in (0, 1)$ denote a threshold for truncating the value.*
- *Let $\alpha \in (0, 1)$ denote a fraction of mass (larger than $\tau$) outside $S_0$.*
- *Let mapping $\mathcal{D} : \mathbb{R}^d \to \mathbb{R}^m_{\geq 0}$.*
- *For each $x \in \mathbb{R}^d$, $\mathcal{D}(x) \in \mathbb{R}^m$ is a vector that has length $m$.*

*We say the distribution $\mathcal{D}$ is $(\alpha, \tau, k)$-good if the following conditions hold*

- *For all $x \in \mathbb{R}^d$, $S_0 \subset \operatorname{supp}_\tau(\mathcal{D}(x))$*
- *For all $x \in \mathbb{R}^d$, $| \operatorname{supp}_\tau(\mathcal{D}(x)) \backslash S_0 | \leq \alpha \cdot k$*

**Claim D.34.** *Suppose we sample $n$ points $\{x_1, x_2, \cdots, x_n\} \subset \mathbb{R}^d$ from $(\alpha, \tau, k)$-good distribution uniformly at random, then we have*

- *$S_0 \subseteq \cap_{i \in [n]} \operatorname{supp}_\tau(x_i)$*
- *$|(\cup_{i \in [n]} \operatorname{supp}_\tau(\mathcal{D}(x))) \backslash S_0| \leq \alpha k n$*

*Proof.* Since for all $i \in [n]$, we have $S_0 \subseteq \operatorname{supp}_\tau(\mathcal{D}(x_i))$, thus
$$S_0 \subseteq \cap_{i \in [n]} \operatorname{supp}_\tau(x_i).$$

Therefore we proved the first property.

We know that for all $i \in [n]$, we have $| \operatorname{supp}_\tau(\mathcal{D}(x)) \backslash S_0 | \leq \alpha k n$. Thus

$$|(\cup_{i \in [n]} \operatorname{supp}_\tau(\mathcal{D}(x_i))) \backslash S_0| \leq \sum_{i=1}^n | \operatorname{supp}_\tau(\mathcal{D}(x_i))) \backslash S_0| \leq n \cdot \alpha k$$

Therefore, we finish the proof for the second property. $\square$

### D.17 Definition of Loss Function

In this section, we follow the theoretical softmax regression literature [105] and define a number of functions to make the calculations of gradient and Hessian convenient. We also proposed a new penalty term ($\ell_1$ type sparsity penalty, see Definition D.41) into the final loss function, which is not studied in previous work [104, 105, 107, 110, 132, 133]. We first provide some function definitions.

**Definition D.35** (Function $u$, [105]). *Given matrix $A \in \mathbb{R}^{n \times d}$, let function $u : \mathbb{R}^d \to \mathbb{R}^n$ be defined as follows*

$$u(x) := \exp(Ax)$$

**Definition D.36** (Function $\alpha$, see Definition 5.4 in [105] as an example). *We define $u(x)$ as Definition D.35. Then we define $\alpha : \mathbb{R}^d \to \mathbb{R}$ as follows*

$$\alpha(x) := \langle u(x), \mathbf{1}_n \rangle$$

**Definition D.37** (Function $f$, see Definition 5.1 in [105] as an example). *Provided that the following conditions are true*

- *We define $u(x)$ as Definition D.35.*

- *We define $\alpha(x)$ as Definition D.36*

*Let function $f : \mathbb{R}^d \to \mathbb{R}^n$ be defined as follows*

$$f(x) := \alpha(x)^{-1} u(x).$$

**Definition D.38** (Function $c$, see Definition 5.5 in [105] as an example). *Provided that the following conditions are true*

- *Given a vector $b \in \mathbb{R}^n$.*

- *Let $f(x)$ be defined as Definition D.38.*

*Then, let function $c : \mathbb{R}^d \to \mathbb{R}^n$ defined as follows*

- *$c(x) := f(x) - b$.*

**Definition D.39** (Loss function $L_{\exp}$, see Definition 5.3 in [105] as an example). *We define $L_{\exp} : \mathbb{R}^d \to \mathbb{R}$*

$$L_{\exp}(x) := 0.5 \cdot \|c(x)\|_2^2.$$

**Definition D.40** (Loss function $L_{\mathrm{reg}}$). *Given $A \in \mathbb{R}^{n \times d}$.*

*Let function $L_{\mathrm{reg}} : \mathbb{R}^d \to \mathbb{R}$ be defined as follows*

$$L_{\mathrm{reg}}(x) := 0.5 \cdot \|\operatorname{diag}(w) Ax\|_2^2$$

We define a novel penalty function

**Definition D.41** (Implicitly controlling the sparsity). *Given $A \in \mathbb{R}^{n \times d}$.*

*We define*

$$L_{\mathrm{sparse}}(x) := \|\exp(Ax)\|_1.$$

Then it is obvious that we have

**Claim D.42.** *Given $A \in \mathbb{R}^{n \times d}$. Let $u$ be defined as Definition D.35. Let $\alpha$ be defined as Definition D.36.*

*We have*

- *$L_{\mathrm{sparse}}(x) = \langle \exp(Ax), \mathbf{1}_n \rangle$*

- *$L_{\mathrm{sparse}}(x) = \langle u(x), \mathbf{1}_n \rangle$*

- $L_{\mathrm{sparse}}(x) = \alpha(x)$

*Proof.* The proof is trivially following from the definition of $u(x)$ (see Definition D.35) and $\alpha(x)$ (see Definition D.36). □

The final loss function can be defined as follows. Intuitively, we can write attention $D^{-1}\exp(QK^\top)$ into $n$ subproblems where each subproblem can be viewed as one softmax problem.

**Definition D.43.** *If the following conditions hold*

- *We define $L_{\mathrm{exp}}$ as Definition D.39.*

- *We define $L_{\mathrm{reg}}$ as Definition D.40.*

- *We define $L_{\mathrm{sparse}}$ as Definition D.41.*

*Then we define $L$ function*

$$L(x) := L_{\mathrm{exp}}(x) + L_{\mathrm{sparse}}(x) + L_{\mathrm{reg}}(x).$$

## D.18 Gradient

Next, we show the gradient of $L_{\mathrm{exp}}$.

**Lemma D.44** (Gradient, Lemma 5.6 in [105]). *Provided that the following conditions are true*

- *Given matrix $A \in \mathbb{R}^{n \times d}$ and a vector $b \in \mathbb{R}^n$.*

- *Let $A_{*,i} \in \mathbb{R}^n$ denote the $i$-th column of matrix $A$, for every $i \in [d]$.*

- *We define $\alpha(x)$ as Definition D.36.*

- *We define $f(x)$ as Definition D.37.*

- *We define $c(x)$ as Definition D.38.*

- *We define $L_{\mathrm{exp}}(x)$ as Definition D.39.*

- *Let $\circ$ denote hadamard product.*

*For every $i \in [d]$, we have*

- *Part 1.*

$$\frac{\mathrm{d}\exp(Ax)}{\mathrm{d}x_i} = \exp(Ax) \circ A_{*,i}$$

- *Part 2.*

$$\frac{\mathrm{d}\langle \exp(Ax), \mathbf{1}_n \rangle}{\mathrm{d}x_i} = \langle \exp(Ax), A_{*,i} \rangle$$

- *Part 3.*

$$\frac{\mathrm{d}\alpha(x)^{-1}}{\mathrm{d}x_i} = -\alpha(x)^{-1} \cdot \langle f(x), A_{*,i} \rangle$$

- *Part 4.*

$$\frac{\mathrm{d}f(x)}{\mathrm{d}x_i} = \frac{\mathrm{d}c(x)}{\mathrm{d}x_i} = -\langle f(x), A_{*,i} \rangle \cdot f(x) + f(x) \circ A_{*,i}$$

- *Part 5.*

$$\frac{\mathrm{d}L_{\mathrm{exp}}(x)}{\mathrm{d}x_i} = A_{*,i}^\top \cdot (f(x)(f(x)-b)^\top f(x) + \mathrm{diag}(f(x))(f(x)-b))$$

### D.19  Hessian

Here, we compute the Hessian for several functions.

**Lemma D.45** (Hessian of $u(x)$, Lemma 5.9 in [105]). *Provided that the following conditions are true*

- *Given a matrix $A \in \mathbb{R}^{n \times d}$.*
- *For every $i \in [d]$, let $A_{*,i} \in \mathbb{R}^n$ denote the $i$-th column of matrix $A$.*
- *Let $\circ$ denote hadamard product.*

*Then, we have, for each $i \in [d]$*

- *Part 1.*
$$\frac{\mathrm{d}^2 \exp(Ax)}{\mathrm{d}x_i^2} = A_{*,i} \circ u(x) \circ A_{*,i}$$

- *Part 2.*
$$\frac{\mathrm{d}^2 \exp(Ax)}{\mathrm{d}x_i \mathrm{d}x_j} = A_{*,j} \circ u(x) \circ A_{*,i}$$

**Lemma D.46** (Lemma 5.10 in [105]). *Provided that the following conditions are true*

- *We define $\alpha(x)$ as Definition D.36.*
- *For every $i \in [d]$, let $A_{*,i} \in \mathbb{R}^n$ denote the $i$-th column of matrix $A$.*
- *Let $\circ$ denote hadamard product.*

*Then, we have*

- *Part 1.*
$$\frac{\mathrm{d}^2 \alpha(x)}{\mathrm{d}x_i^2} = \langle u(x), A_{*,i} \circ A_{*,i} \rangle$$

- *Part 2.*
$$\frac{\mathrm{d}^2 \alpha(x)}{\mathrm{d}x_i \mathrm{d}x_j} = \langle u(x), A_{*,i} \circ A_{*,j} \rangle$$

- *Part 3.*
$$\frac{\mathrm{d}^2 \alpha(x)}{\mathrm{d}x^2} = A^\top \operatorname{diag}(u(x)) A$$

### D.20  Hessian is Positive Definite

It is well known that in literature [104, 105, 107], the Hessian $H$ of loss function can be written as $A^\top(B(x) + W^2)A$ for some matrix function $B(x) \in \mathbb{R}^{n \times n}$ (for example see explanation in Section 5.10 in [105]). In this section, we show that Hessian is positive definite.

**Lemma D.47.** *If the following conditions hold*

- *Given matrix $A \in \mathbb{R}^{n \times d}$.*
- *We define $L_{\mathrm{sparse}}(x)$ as Definition D.41.*
- *We define $L_{\mathrm{sparse}}(x)$ Definition D.40.*
- *We define $L_{\mathrm{sparse}}(x)$ Definition D.39.*
- *Let $L(x) = L_{\exp}(x) + L_{\mathrm{sparse}}(x) + L_{\mathrm{reg}}(x)$.*

- *Let $A^\top(B(x) + W^2)A$ be the Hessian of $L(x)$*

- *Let $W = \text{diag}(w) \in \mathbb{R}^{n \times n}$. Let $W^2 \in \mathbb{R}^{n \times n}$ denote the matrix that $i$-th diagonal entry is $w_{i,i}^2$.*

- *Let $\sigma_{\min}(A)$ denote the minimum singular value of $A$.*

- *Let $l > 0$ denote a scalar.*

*Then, we have*

- *Part 1. If all $i \in [n]$, $w_i^2 \geq 20 + l/\sigma_{\min}(A)^2$, then*

$$\frac{\mathrm{d}^2 L}{\mathrm{d}x^2} \succeq l \cdot I_d$$

- *Part 2 If all $i \in [n]$, $w_i^2 \geq 200 \cdot \exp(R^2) + l/\sigma_{\min}(A)^2$, then*

$$(1 - 1/10) \cdot (B(x) + W^2) \preceq W^2 \preceq (1 + 1/10) \cdot (B(x) + W^2)$$

*Proof.* The entire proof framework follows from [104, 105, 107], in the next few paragraphs, we mainly explain the difference.

The $B(x)$ based on $L_{\mathrm{sparse}}$ is $\mathrm{diag}(u(x))$. Note that it is obvious that

$$\mathrm{diag}(u(x)) \succeq 0.$$

From the upper bound size, we know that

$$\mathrm{diag}(u(x)) \preceq \|u(x)\|_\infty \cdot I_n$$
$$\preceq \exp(R^2)$$

where the last step follows from Proof of Part 0 in Lemma 7.2 in [105].

To prove Part 1, following from [104, 105], we only use the lower bound of $\mathrm{diag}(u(x))$. By putting things together, we get our results.

To prove Part 2, we follow from [107] and use both the upper bound and lower bound of $\mathrm{diag}(u(x))$.

$\square$

### D.21 Hessian is Lipschitz

In this section, we show Hessian is Lipschitz.

**Lemma D.48.** *If the following conditions hold*

- *Let $H(x) = A^\top(B(x) + W^2)A$ denote the Hessian of L.*

*Then, we have*

- $\|H(x) - H(y)\| \leq n^2 \exp(40R^2)\|x - y\|_2$

*Proof.* The entire proof framework follows from [104, 105, 107], in the next few paragraphs, we mainly explain the difference.

Note that the $B(x)$ based on $L_{\exp} + L_{\mathrm{reg}}$ have been proved by [105]

We only need to prove $B(x)$ based on $L_{\mathrm{sparse}}$ and add them together.

Note that $B(x)$ based on $L_{\mathrm{sparse}}$ is in fact $\mathrm{diag}(u(x))$.

Using Lemma 7.2 in [105], we know that

$$\|\mathrm{diag}(u(x)) - \mathrm{diag}(u(y))\| \leq \|u(x) - u(y)\|_2$$
$$\leq 2\sqrt{n}R\exp(R^2)\|x - y\|_2$$

where the last step follows from Part 1 in Lemma 7.2 in [105].

Thus, putting things together, we complete the proof.

$\square$

### D.22 Greedy Type Algorithm

In this section, we propose a greedy-type algorithm (based on the approximate Newton method) to solve the optimization problem.

---

**Algorithm 4** A greedy type algorithm.

---

1: **procedure** OURITERATIVEMETHOD($A \in \mathbb{R}^{n \times d}, b \in \mathbb{R}^n, w \in \mathbb{R}^n, \epsilon, \delta$)
2:     Initialize $x_0$
3:     $T \leftarrow \log(\|x_0 - x^*\|_2 / \epsilon)$                        $\triangleright$ Let $T$ denote the number of iterations.
4:     **for** $t = 0 \rightarrow T$ **do**
5:         $D \leftarrow B_{\mathrm{diag}}(x_t) + \mathrm{diag}(w \circ w)$
6:         $\widetilde{D} \leftarrow$ SUBSAMPLE$(D, A, \epsilon_1 = \Theta(1), \delta_1 = \delta/T)$
7:         Compute gradient $g$ exactly
8:         Get the approximate Hessian $\widetilde{H}$ by computing $A^\top \widetilde{D} A$
9:         Update $x_{t+1}$ by using the Newton step $x_t + \widetilde{H}^{-1} g$
10:    **end for**
11:    $\widetilde{x} \leftarrow x_{T+1}$
12:    **return** $\widetilde{x}$
13: **end procedure**

---

**Theorem D.49.** *Given matrix $A \in \mathbb{R}^{n \times d}$, $b \in \mathbb{R}^n$, and $w \in \mathbb{R}^n$.*

- *We use $x^*$ to denote the optimal solution of*

$$\min_{x \in \mathbb{R}^d} L_{\mathrm{exp}} + L_{\mathrm{sparse}} + L_{\mathrm{reg}}$$

   *that*

     – *$g(x^*) = \mathbf{0}_d$, where $g$ denotes the gradient function.*
     – *$\|x^*\|_2 \leq R$.*

- *Suppose that $R \geq 10$, $M = \exp(\Theta(R^2 + \log n))$, and $l > 0$.*

- *Assume that $\|A\| \leq R$. Here $\|A\|$ denotes the spectral norm of matrix $A$.*

- *Suppose that $b \geq \mathbf{0}_n$ and $\|b\|_1 \leq 1$. Here $\mathbf{0}_n$ denotes a length-$n$ vector where all the entries are zeros. (Here $b \geq \mathbf{0}_n$ denotes $b_i \geq 0$ for all $i \in [n]$)*

- *Assume that $w_i^2 \geq 200 \cdot \exp(R^2) + l/\sigma_{\min}(A)^2$ for all $i \in [n]$. Here $\sigma_{\min}(A)$ denotes the smallest singular value of matrix $A$.*

- *Let $x_0$ denote an starting/initial point such that $M\|x_0 - x^*\|_2 \leq 0.1l$.*

- *We use to $\epsilon \in (0, 0.1)$ represent our accuracy parameter.*

- *We use $\delta \in (0, 0.1)$ to represent failure probability.*

*There is a randomized algorithm that*

- *runs $\log(\|x_0 - x^*\|_2 / \epsilon)$ iterations*

- *spend*

$$O((\mathrm{nnz}(A) + d^\omega) \cdot \mathrm{poly}(\log(n/\delta))$$

   *time per iteration,*

- *and finally outputs a vector $\widetilde{x} \in \mathbb{R}^d$ such that*

$$\Pr[\|\widetilde{x} - x^*\|_2 \leq \epsilon] \geq 1 - \delta.$$

*Proof.* The proof framework follows from approximate Newton (second order method) literature [134, 135, 136, 137, 138, 139, 140, 141, 104, 105, 107, 110, 111, 142, 143, 144, 145].

Following from Lemma D.47, we know the Hessian of the loss function is positive definite.

Following from Lemma D.48, we know the Hessian of $L$ is Lipschitz.

Following Section 9 in [105], by running Algorithm 4, we complete the proof.

$\square$

We remark that $\omega$ denotes the exponent of matrix multiplication (i.e., $n^\omega$ is the time of multiplying an $n \times n$ matrix with another $n \times n$ matrix). The most naive algorithm gives $\omega = 3$. Currently, the state-of-the-art algorithm gives $\omega = 2.373$.

