# OpenReview forum: "H2O: Heavy-Hitter Oracle for Efficient Generative Inference of Large Language Models"
_NeurIPS.cc/2023/Conference — NeurIPS 2023 poster_

### Official Review · Reviewer_nR7P · 2023-07-06

**Soundness:** 3 good
**Presentation:** 3 good
**Contribution:** 3 good
**Rating:** 6
**Confidence:** 4

**Summary:**

The author proposes to reduce the size of KV cache to accelerate the inference of the Transformer models. The size reduction is conducted through a mechanism called heavy hitter oracle, which treats attention score as the importance score. The authors conduct experiments with various tasks and observe consistent improvements.

**Strengths:**

The studied problem is important. The proposed method is simple. The improved throughput in Section 5.2 seems significant. The theoretical analyses provide some bonus on the greedy algorithm.

**Weaknesses:**

The submission lacks several crucial empirical evidence to support the efficiency claim (i.e., 'reduce KV cache memory footprint by 5-10x while achieving comparable accuracy on a majority of tasks').

My major concern for this study is the gap between the evaluation in 5.1 and 5.2. Specifically, the evaluation conducted in Section 5.2 is on the synthetic/toy data, while the major results in Section 5.1 are evaluated on real-world datasets. This inconsistency left doubt on what would be the throughput improvement of the proposed method on real-world datasets. Moreover, if the throughput improvement on real-world datasets is not as significant as in section 5.2, the results in section 5.2 would be a bit misleading.

Furthermore, the baseline considered in the experiments appears too simplistic. Comparisons with the simple Top-K baseline are only briefly discussed in the appendix. The main body of the submission only contains comparisons with a very naive baseline called Local. Since It is observed that some attention heads only attend to the first token or special token, I think the bad performance of Local is as expected. Since most experiments are conducted in a few-shot setting, I recommend directly using 1-shot or 0-shot full models as baselines. Note that this approach differs from simply dropping the cache for middle tokens, as Transformers utilize positional encoding.

Also, for experiments in Section 5.1, I suggest also to report the zero/one-shot performance. Otherwise, it is unclear whether the proposed method only applies to scenarios with lots of redundancy on model inputs (e.g., 5-shot in-context learning).

**Questions:**

What is the reason for emitting the throughput improvements for experiments in Section 5.1?

**Limitations:**

As the key focus of this paper is efficiency improvement, elaborations on this aspect would significantly improve the quality of this study. For example, in Table 2, I would recommend reporting the throughput of H2O under both GPU offloading and CPU offloading.

---

> ### Author Rebuttal · Authors · 2023-08-10
>
> We thank the reviewer for really insightful and detailed questions and suggestions. These help us strengthen experimental results and make more precise claims. We present details of our implementation as well as additional experiment results that we have added to the updated draft below.
>
> **[Q1: Throughput on Real-World Datasets]** This is a great catch! Firstly, we’d like to clarify the reason why we comprehensively assess the throughput improvements of our method with different simulated  sequence length/batch size in a separate section (section 5.2).
> - First, the throughput improvements of H$_2$O depend mainly on the sequence length of input examples, the length of the generated text, and the predefined KV cache budget, hardware spec, but do not depend on the specific content of the data. Since it is hard to find benchmarks that cover all the cases, especially the popular long-context generation which is known to be very difficult to evaluate, we use simulated data to show different system performance in a wide-range of settings.
> - Second, independent of the eviction policy proposed in our current work, section 5.2 also aims to provide a good evaluation on our system implementation which can support more future work on different or even better policies. For example, this interesting work [1] that shows up after the submission deadline presents a trainable KV cache compression policy and can potentially be deployed in our system with slight modification on the policy.
>
> **Results**: However, for consistency and precision of scientific claims, we agree that it is very necessary to provide the throughput improvement on real-world datasets. As shown in Table R1, with our KV cache eviction policy, the overall throughput is improved by up to 3x, 29x, and 22x over FlexGen, Accelerate, and DeepSpeed, respectively. The results demonstrate that our method can provide significant throughput improvement in real-world applications. And we will include more extensive results for the real benchmarks and revise our claims in the updated version.
>
> Table R1: Results of generation throughput (token/s) on a T4 GPU with different systems on real-world datasets, XSUM. The number in the bracket denotes the effective batch size and the lowest level of the memory hierarchy that the system needs for offloading, where “C” means CPU and “G” means GPU.
>
> | Model size | 6.7B | 30B |
> | --- | --- | --- |
> |Accelerate |11.98 (1, G)| 0.23 (2, C) |
> |DeepSpeed|3.52 (6, C) | 0.31 (2, C) |
> | FlexGen |10.80 (1, G)  | 3.29 (44, C) |
> | H2O (20%) | 30.40 (1, G) | 6.70 (180, C) |
>
> [1]  Anagnostidisµ et al. Dynamic Context Pruning for Efficient and Interpretable Autoregressive Transformers.
>
> **[Q2: More Competitive Baselines]** Thanks for helping us come up with this solid baseline. We have conducted additional experiments of zero/one-shot inference with full cache size and use it as a more competitive baseline.
>
> **Results**: We show in Table R2 that H$_2$O consistently outperforms both the zero-shot and one-shot baselines, with performance enhancements of up to $9%$ across diverse tasks.  When H$_2$0 employs a 20\% KV cache for 5-shot inference, it approximates using $1.2$ samples per input, while the zero-shot and one-shot full model use $1$ and $2$ samples, respectively. The observed performance gain further validates the effectiveness of our approach.
>
> Table R2: Results of LLaMA-7B across different tasks. Experiments are conducted under 20\% KV cache budget under the 5-shot inference scenario, except for the two baselines methods (0-shot/1-shot full model)
>
> ​​|PiQA|COPA|OpenbookQA|Winogrande|
> |---|---|---|---|---|
> |Full| 80.09 | 81.00 | 44.80 | 71.51|
> |0-shot Full| 78.89 | 76.00 | 41.40 | 70.00 |
> |1-shot Full|79.11|76.00|43.60|70.24|
> |Local|57.94|56.00|28.40|51.30|
> |Ours|79.22|85.00|43.80|71.67|
>
>
>
>
>
> Furthermore, we have also conducted a comparative analysis between our H$_2$O and another competitive baseline, namely, sparse transformer, detailed in Section 5.1, with results reported in Table 1. Both variants of the sparse transformer method (“strided” and “fixed”) encounter a notable performance degradation of up to $35$% compared to the full cache scenario.
>
>
> **[Q3: Additional Experiments of One/Zero-Shot Inference]** Thanks for pointing this out! We want to clarify that our original incentive of using 5-shot (long context) is simulating the setting that KV-cache memory becomes the bottleneck. However, we agree that there is more redundancy in 5-shots and thus conduct additional experiments to assess our approach under zero/one shot scenarios.
>
> **Results**: Figure R2 of the uploaded PDF shows that H$_2$O can successfully mitigate the memory requirements of the KV cache by up to $5\times$ under both one/zero-shot inference (COPA, LLaMA-7B, one-shot inference; MathQA, LLaMA-7B, zero-shot inference), achieving matching performance as the model with full KV cache while the local method fails. However, unlike 5-shot, we also found that some tasks are more difficult and the model requires more information to generate the correct content, resulting in a higher KV cache budget (30-40%). This might be expected since 0/1 shots in our benchmarks have shorter sequence length (100-300).
>
> In general, H$_2$O achieves consistent effectiveness across various tasks. And we have incorporated all the results in updated manuscripts.
>
>
>
>
> **[Q4: Throughput of H2O under both GPU/CPU Offloading]**
> We have tested on both pure GPU and GPU/CPU offloading in Table 2 (30B models are run with CPU offloading). We will clarify this setting in the updated manuscript to make it easy to follow.
>
> Thanks for reading our responses! Please let us know if the above addresses the questions and concerns.

---

> > ### Comment · Reviewer_nR7P · 2023-08-14
> > **Rebuttal Response**
> >
> > Thanks for the response and please find some follow-up questions are as below:
> > 1. I'm wondering whether the implementation would be released to the public, and also whether the author can provide the implementation for review. If not, I'm wondering whether the author can confirm that Table R1 and Figure 4 are the evaluation of the exact same implementation.
> > 2. Thanks for providing the additional results on XSUM. I'm wondering whether the author can also provide the results on other datasets that used in Section 5.1, i.e., COPA, OpenBookQA, PiQA, Winogrande, etc.
> > 3. I'm wondering how the speedup is measured. I understand that the study here focuses on the generation speed, but I'm whether the time for constructing the H2O cache is included in the generation speed. Intuitively, with the proposed H2O cache, the construction of the KV cache could be more expensive. I'm wondering whether such computation overhead is included in the speed evaluation.

---

> > > ### Author Response · Authors · 2023-08-18
> > > **Responses to follow-up questions**
> > >
> > > Thanks for the follow-up questions and apologies for the late reply. The point-wise responses are provided in the following:
> > >
> > > **[Q1: Implementation Details]** Thanks for the question! We can confirm that all the results(both throughput and performance) of OPT models are based on exactly the same implementation, which is built on top of FlexGen (as mentioned in Line 279-285: “We implement our KV cache eviction policy in a state-of-the-art inference engine, FlexGen, and report
> > > the throughput and latency improvements”). However, FlexGen currently only supports OPT-family models and is still under development for more architectures, thus we benchmark the accuracy of our methods on other architectures, including LLaMA and GPT-NeoX based on the implementation of HuggingFace. We’ve already made the details more clear in the updated version of our manuscript. Meanwhile, we have contacted FlexGen authors to support more architectures and will release the corresponding implementation of H$_2$O in the future. While we are not allowed to post anonymous code links during the discussion phase, all the implementation along with the additional results during the rebuttal period will definitely be open-sourced.
> > >
> > > **[Q2: Results on Classification Tasks]** This is a great question. The classification tasks like COPA, OpenBookQA, PiQA, and Winogrande only require one-step generation, and thus generation efficiency is not the bottleneck. However, similar to other efficient generation work[1-3], we use these tasks to evaluate the performance or accuracy of our KV cache eviction policy. We simulate these tasks as if every token of the prompt is generated sequentially, and then apply our KV cache eviction policy when the number of KV embeddings reaches the cache budget. Furthermore, we evaluate both the performance and throughput on real-world generation tasks like XSUM and demonstrate consistent throughput improvement compared with other inference systems while preserving the performance with a full KV cache.
> > >
> > > Ideally, since our work aims to improve the generation efficiency, e.g., key bottleneck for applications like ChatGPT, we would like to evaluate it more in those settings. However, benchmarking long content generation is still an open problem, we’d appreciate it if the reviewer could provide some suggestions about long content generation tasks and we are happy to add more evaluation to support our methods.
> > >
> > > [1] Liu et al. Deja Vu: Contextual Sparsity for Efficient LLMs at Inference Time.
> > >
> > > [2] Sheng et al. FlexGen: High-Throughput Generative Inference of Large Language Models with a Single GPU.
> > >
> > > [3]Frantar et al. GPTQ: ACCURATE POST-TRAINING QUANTIZATION FOR GENERATIVE PRE-TRAINED TRANSFORMERS
> > >
> > >
> > >
> > > **[Q3: Details about SpeedUp Measurement]** All the speedup results are tested in an end-to-end setting, including pre-filling and generation phase. It includes the time for constructing the H$_2$O KV cache.

---

> > > > ### Comment · Reviewer_nR7P · 2023-08-18
> > > > **Response Follow-up**
> > > >
> > > > Thanks for the prompt response. The major reason that I try to be more careful about this submission is that during reading the paper, I was alerted by some details. For example, I am aware that FlexGen only supports OPT, and this alerts me when I saw the authors conduct experiments on LLaMA and mentioned *"We implement H2O on top of FlexGen which can easily adapt different cache eviction techniques to produce a system with high-throughput inference"*. As these concerns are confirmed by the author, I am asking more follow-up questions.
> > > >
> > > > Re-Q1: I appreciate your acknowledgment of the discrepancy between the OPT-family experiments and other experiments. **Such crucial details should be discussed in the manuscript**, given that accurate implementation details significantly influence the success of deep learning. Regarding code sharing, the NeurIPS rebuttal guideline mentions: "If you were asked by the reviewers to provide code, please send an anonymized link to the AC in a separate comment (ensuring the code and all associated files and filenames are completely anonymized)." **Please consider this if feasible**.
> > > >
> > > > Re-Q2: Given the study's nature (the proposed method only works for the generation task), I strongly advise a different experimental design. Specifically, as the efficiency evaluation is conducted on generating **32/512/1024 tokens**, the quality of the generated texts should also be evaluated on generation tasks in a similar setting. It is not suitable and not sufficient to evaluate generated texts mostly on tasks that only require **one token** to be generated, for reasons elaborated below.
> > > >
> > > > One core motivation of this study is that the eviction w. global statistic strategy is not feasible in typical generation settings (as in Figure 3) and can be approximated by local statistics. Accordingly, comparing the global statistic and the local statistic is very crucial to verify the significance of both the empirical contribution and the theoretical contribution (Theorem 4.4).
> > > > However, in classification tasks (i.e., COPA, OpenBookQA, PiQA, and Winogrande), since only one-step generation is needed, the difference between their local statics and their global statics **cannot represent** the general case while multiple generation steps are needed. The author should discuss this in the submission, instead of only using single-step generation tasks in Figure 2 (d).
> > > >
> > > > Also, *"similar to other efficient generation work[1-3]"* cannot justify this evaluation setting. On one hand, these methods have different natures compared to the submission, and they do not attempt to approximate the global statistic with the local statistic. For example, the quantization method [3] does employ the greedy algorithm as discussed in Section 4.1, and thus the one-step generation is a reasonable testbed for this method. On the other hand, these methods are not listed as baselines in this study. As there are no comparisons with these methods (I agree there is no need to make such comparisons), there is no need to follow their evaluation settings.
> > > >
> > > > I would suggest the authors conduct experiments on more generation tasks like 1. using AlpacaEval to evaluate instruction-following language models; 2. using generation tasks as the benchmark to evaluate the generated texts (e.g., human-eval, mbpp, zero-shot translation).
> > > >
> > > > Re-Q3: Thanks for the confirmation.

---

> > > > > ### Author Response · Authors · 2023-08-21
> > > > > **Responses for Further Questions.**
> > > > >
> > > > > Thanks for the questions. Here are the pointwise responses:
> > > > >
> > > > > **[Implementations]** Thanks for the information, we’ve uploaded the code to AC through an anonymous link.
> > > > >
> > > > > **[Experiment Design and More Results]** Firstly, we’d like to emphasize that the accuracy evaluation of our method on classification tasks is **reasonable**. We have mentioned in the previous message to the reviewer that despite the fact that classification tasks only need one-step generation, we evaluate their accuracy using **multi-step generations**, as if every token of the prompt is generated sequentially after we reach the cache budget. For example, if the budget is 20 tokens and the input prompt has 40 tokens, the computation of every kv after the 22st token will be approximate, not exact since it is supposed to see only 20 of them. In this way, we still require the greedy algorithm based on **local statistics** to approximate global statistics, otherwise causing performance degradation.
> > > > >
> > > > > In addition, we mentioned [1-3] because similar to language modeling tasks, these classification tasks are standard benchmarks to evaluate how much we lose in accuracy and quality when involving approximation in LLM generation steps, although throughput evaluation does not apply in those settings. **Specifically, since the reviewer mentioned [3], if we could see table 6 (page 8), the speed number is also evaluated in simulated** **generation length of 128 while the tasks are all one or few token generation or classification tasks**. **[1,2] and many more in LLM** **approximate generation literature are** **similar.**
> > > > >
> > > > > Again, it is easy to perform evaluation on the same tasks for both accuracy and speed if the work is targeted on prompting/pre-filling phase, but very challenging for generation. H$_2$O has already had more solid evaluations compared to the literature on real **generation tasks**, like XSUM and CNN/Daily Mail. But we want to thank the reviewer for the detailed suggestions on further evaluations to push our work to the next level. We conducted additional experiments on **AlpacaEval** and **MT-bench**. Given AlpacaEval’s increased difficulty relative to previous tasks, we choose to evaluate our approach with a 30% KV cache budget and leveraged GPT-4 to compare the quality of generated answers from both our H$_2$O and Full models. We could only evaluate with limited hyperparameter and models due to the limited time, computational resources, and each submission costs **$10$** dollars. The results indicate a competitive performance of $H_2O$ compared to the full dense model where the winning rate of our method reaches **$49.00$** and **$47.45$** for OPT-IML-1.3B and Llama-2-chat-7B, respectively.
> > > > >
> > > > > For MT-bench, the instruction sequence length is larger (500-1100) than that in AlpacaEval, rendering the KV cache size a more pronounced limiting factor during generation. Consequently, we opted for a more assertive 20% KV cache budget for our evaluation. As shown in Table R3, our approach achieves a **similar** performance of the full model in both turns. It's worth noting that due to time constraints and the associated costs of evaluating outputs with GPT-4, we couldn't thoroughly tune all hyperparameters, which may cause our results to be sub-optimal. But we believe these results can further validate the effectiveness of our method and we’ve incorporated them into our updated version.
> > > > >
> > > > > Table R3: Results on MT-bench of Llama-2-chat-7b.
> > > > > |Score|First Turn|Second Turn|Average|
> > > > > |---|---|---|---|
> > > > > |Full|6.8688|5.9438|6.4063|
> > > > > |Ours (20%)|6.7500|5.7750|6.2625
> > > > >
> > > > > We hope the above explanation and additional experiments can address the concerns and make the reviewer reconsider the recommendation to our work. Thanks!
> > > > >
> > > > >
> > > > > [1] Liu et al. Deja Vu: Contextual Sparsity for Efficient LLMs at Inference Time.
> > > > >
> > > > > [2] Sheng et al. FlexGen: High-Throughput Generative Inference of Large Language Models with a Single GPU.
> > > > >
> > > > > [3] Frantar et al. GPTQ: ACCURATE POST-TRAINING QUANTIZATION FOR GENERATIVE PRE-TRAINED TRANSFORMERS

---

### Official Review · Reviewer_N7jZ · 2023-07-07

**Soundness:** 4 excellent
**Presentation:** 4 excellent
**Contribution:** 4 excellent
**Rating:** 8
**Confidence:** 4

**Summary:**

The paper presents H2O, a technique to identify key elements in the sequence whose KV values play the most important role in attention scores for generation of future tokens and proposes to only keep these in cache to improve upon memory requirement and latency during inference.

The proposed algorithm is then formulated as a dynamic submodular problem through which theoretical guarantees are provided for the effectiveness of the algorithm.

**Strengths:**

The algorithm in itself is pretty impactful. Given the amount of work done in trying to improve the efficiency of serving large language models, this algorithm presents a pretty impactful way to retain most of the quality while increasing memory requirement, throughput and latency by huge margins.

The proposed algorithm is well analyzed through theoretical guarantees, experiments, and ablations.

The paper explores an important problem that has huge impact on the usability of large models by making them less resource intensive.

**Weaknesses:**

- Some of the equations in the paper are difficult to follow because of the choice of notations. Especially, [i] and 1_{[i]}
- In Algorithm 1, line 9, should the tokens that are already discarded from the cache ([i]/S_{[i]}) be masked out similar to what has been done on line 8?
- In Algorithm 1, K_{S_{i}} is used in the equations (line 8 and 9), while S_{i} is initialized only at line 13.

**Questions:**

- Do you have any analysis/thought that incorporates this during training as well thereby removing the inference & training skew?
- Do you have any comparison with similar alternative methods like ColT5? https://arxiv.org/abs/2303.09752

---

> ### Author Rebuttal · Authors · 2023-08-10
>
> Many thanks to the reviewer for acknowledging that the problem we explored is “important”, “has huge impact”, and also our algorithm is “impactful”. We provide detailed responses below to address the constructive feedback.
>
> **[Q1: Inconsistent Notations]** Thanks for pointing it out. To fix the inconsistent notations, we have modified line 8, 9 of Algorithm 1 in the updated manuscript.
>
> **[Q2: Comparison with Alternative Methods]** Thanks for the question.
> CoLT5 and H$_2$O address a similar problem but with different approaches. We first present positive results on a preliminary study of combining H$_2$O with the backbone of CoLT5, i.e., T5, which suggests that although densely trained, similar to auto-regressive models, T5 also has sparse structures at inference time. This explains why it precisely supports why CoLT5 can improve training and inference speed by routing tokens to different branches.  Finally, we conclude with a discussion of the main difference between CoLT5 and our approach.
>
> **Results**: As shown in Table R1, our approach can preserve the performance with only a 20% KV cache budget across various sizes of T5 models.
>
> Table R1: Results of T5 on the XSUM task.
> |Rouge-L|T5-Large|T5-3B|T5-11B|
> |---|---|---|---|
> |Full|14.6831|14.8753|15.7553|
> |Ours (20%)|14.0567|14.8074|15.5190
>
>
>
> **Discussion**: CoLT5 employs a token-wise conditional computation strategy to reduce the overall computation cost, which requires training the model from scratch while our H$_2$O can be directly plugged into pretrained models without retraining. Additionally, CoLT5 uses a routing network to predict which tokens are more important and need to be processed by the heavy branch. But our approach leverages a simple but effective heuristic (accumulated attention scores) to identify the “heavy-hitter” tokens at inference time.
>
> Furthermore, our method can naturally be incorporated with CoLT5 to further reduce the KV cache requirement in its heavy branch. Since there aren’t open-sourced pretrained CoLT5 models and it’s too difficult to train one model on our own in such a short period of time, we only present some preliminary results of applying H$_2$O on the backbone of CoLT5 to demonstrate how our method can be introduced in the heavy branch of CoLT5 above. We will add the reference and conduct the experiment of applying H$_2$O on CoLT5 in the future once the pre-trained CoLT5 models are released.
>
> [1] COLT5: Faster Long-Range Transformers with Conditional Computation.
>
> **[Q3: Apply H$_2$O into Training]** Thanks for the good and natural question! We conduct an additional experiment of applying H$_2$O into training and observe positive results.
>
> **Results**: After training from scratch, the resulting model from our approach reaches 67.68 Perplexity (PPL) on the validation set, while the baseline model only shows 69.02 PPL. The performance improvement demonstrates the possibility of introducing H$_2$O into training and we will comprehensively study it in the future.
>
> **Details**: Based on the results shown in Figure 2(b), heavy-hitter tokens have a high co-occurrence property in the data. With this property, we first calculate the co-occurrences statistic in the training data and identify the heavy-hitter tokens. Then, we manually modify the attention scores during training to encourage the model to focus more on the heavy-hitter tokens. Experiments are implemented with a GPT-2 model on 10% training data from Wiki-Text-103.
>
> Thanks for reading our responses! Please let us know if the above addresses the questions and concerns.

---

> > ### Comment · Reviewer_N7jZ · 2023-08-16
> >
> > Thanks for the response. Read through it and will continue with my original scores.

---

> > > ### Author Response · Authors · 2023-08-18
> > > **Thanks for the response**
> > >
> > > Dear Reviewer N7jZ,
> > >
> > > We sincerely appreciate all the constructive feedback and very positive evaluations from reviewer N7jZ.
> > >
> > > Thanks again for your time and support!
> > >
> > > Best wishes,
> > >
> > > Authors

---

### Official Review · Reviewer_CXTK · 2023-07-21

**Soundness:** 4 excellent
**Presentation:** 4 excellent
**Contribution:** 4 excellent
**Rating:** 8
**Confidence:** 4

**Summary:**

This paper first identified an interesting observation that only a small portion of tokens contributes most of
the value when computing attention scores. And based on this observation, this paper proposed Heavy Hitter Oracle (H2O), a KV cache eviction policy that dynamically retains a balance of recent and H2 tokens. The results show significant improvement in memory reduction and inference speedup.


**Strengths:**

- The paper is well-written and easy to follow.
- Strong performance in terms of decoding latency and memory reduction.
- The idea of evicting $(K,V)$ of non-important tokens from the cache is intuitive and works well.

**Weaknesses:**

Minor:
- I wonder what would happen if the current approach is applied during training, although it might be slow. Will it result in better performance since the model can have a better inductive bias to learn from important tokens?
- Would be interesting if we can see the same method work in the encoder-decoder framework.
- I wonder if the same approach can be applied to reduce the memory size in Memorizing Transformer ([link](https://arxiv.org/pdf/2203.08913.pdf)), or essentially any similar soft external memory mechanism in LLM.


**Questions:**

See weakness.

**Limitations:**

yes

---

> ### Author Rebuttal · Authors · 2023-08-10
>
> We thank the reviewer for strongly supporting our work and providing valuable feedback, which helps improve our work. We present pointwise responses for your constructive suggestions as the following:
>
>
> [**Q1: Results on Encoder-Decoder Framework**] Thanks for the suggestion. We further evaluate our method with representative encoder-decoder models, including T5-Large, T5-3B, and T5-11B.
>
> **Results**: As shown in Table R1, with a 20% KV cache budget, H$_2$O can match the performance of the full model across different model sizes, which demonstrates our approach is also effective in the Encoder-Decoder framework.
>
> Table R1: Results of T5 models on the XSUM task.
> |Rouge-L|T5-Large|T5-3B|T5-11B|
> |---|---|---|---|
> |Full|14.6831|14.8753|15.7553|
> |Ours (20%)|14.0567|14.8074|15.5190
>
> **Details**: Experiments are conducted on the XSUM task, and we use the Rouge-L score to evaluate the model’s performance.
>
>
> [**Q2: Reduce the Memory Size in Memorizing Transformer**] This is a good suggestion. We investigate the potential applications of our approach for improving memorizing transformers.
>
> **Results**:  The results are reported in Table R2. With the same memory size, the heavy-hitter strategy can effectively improve the final performance while restoring only the non-heavy-hitter or random tokens causing performance degradation. This positive result demonstrates the possibility of reducing the external memory size of memorizing transformers by introducing heavy-hitter tokens. And we will systematically study it in future work.
>
> Table R2: Results of memorizing transformers with heavy-hitter tokens.
> ||Baselines|w. Heavy-Hitters|w. Non-Heavy-Hitters|w. Random Tokens|
> |---|---|---|---|---|
> |Perplexity|4.014|3.962|4.178|4.049|
>
> **Details**: Experiments are conducted with 8-layer memorizing transformers on enwik8 dataset. Four methods are compared, including the vanilla memorizing transformers (baseline), with only heavy-hitter, non-heavy-hitter or random tokens.
>
>
>
> [**Q3: Apply H2O during Training**] This is indeed a natural question! We conducted a preliminary investigation about incorporating heavy-hitter into training.
>
> **Results**: With our approach, the final model reaches 67.68 Perplexity (PPL) on the validation set, while the baseline model only reaches 69.02 PPL. The performance improvement demonstrates the potential of applying our approach during training, and we will investigate more in the future.
>
> **Details**: As illustrated in Figure 2(b),  heavy-hitter tokens exhibit high co-occurrences within the dataset. Based on that, we first calculate the co-occurrences statistic in the training data and identify the heavy-hitter tokens. Then, we manually increase the attention scores of heavy-hitter tokens, encouraging the model to focus more on the heavy-hitter tokens. Experiments are conducted with a GPT-2 model. And we train the model from scratch on 10% data from Wiki-Text-103.
>
> Thanks for reading our responses! Please let us know if the above addresses the questions and concerns.

---

> > ### Comment · Reviewer_CXTK · 2023-08-16
> > **Rebuttal Response**
> >
> > Thank you for the detailed responses. Looking forward to your future work!

---

> > > ### Author Response · Authors · 2023-08-18
> > > **Thanks for the response**
> > >
> > > Dear Reviewer CXTK,
> > >
> > > Many thanks for the helpful comments and positive evaluations from reviewer CXTK. We really appreciate reviewer CXTK for the time and effort during reviewing. Thanks!
> > >
> > > Sincerely,
> > >
> > > Authors

---

### Official Review · Reviewer_AZfb · 2023-07-26

**Soundness:** 3 good
**Presentation:** 2 fair
**Contribution:** 3 good
**Rating:** 7
**Confidence:** 3

**Summary:**

Paper addresses inefficiency in key-value matrix caching during attention computations of generation with transformer models by introducing a new pruning strategy to discard less useful information without significant degradation of performance. This strategy is based on the reliance of transformers' on observable token cooccurrence patterns, where only a subset of tokens significantly contribute to performance. They frame the problem as dynamic submodular optimization, and show empirically that the strategy considerably reduces memory overhead and inference time while preserving most of the original performance.

**Strengths:**

- The authors introduce a simple, intuitive approach that is empirically demonstrated to preserve performance in the zero-shot setting across model architectures/benchmarks. Most of the paper's claims are well-validated.

- They tackle an important issue of LLM inference efficiency (both memory and speed-up), and their approach has many potentially critical practical implications for research engineering work involving LLMs.

**Weaknesses:**

- Given the effort put into the rest of the work, it is disappointing that the authors' claim in 5.3 that H20 leads to more diverse generations is only weakly substantiated through a small qualitative evaluation (assessed by the authors). This should be demonstrated with a large human study or through diversity metrics over an evaluation set (e.g. unique ngrams, self-bleu).

- While they test more model architectures/datasets for the final results, the sparsity findings are only considered for OPT model generations. This is also reported in isolation, without reference to prior work on sparsity in attention matrices, even though this has been widely studied, and gives the impression that the authors are claiming this as a novel observation and sole motivation for their "heavy-hitters" strategy. I would suggest adding citations to previous work noting sparsity, e.g:

1. Edelman, B., Goel, S., Kakade, S.M., & Zhang, C. 2022. Inductive Biases and Variable Creation in Self-Attention Mechanisms. ICML.
2. Likhosherstov, V., Choromanski, K., & Weller, A. 2021. On the Expressive Power of Self-Attention Matrices. ArXiv, abs/2106.03764.

**Questions:**

Presumably this approach is implemented uniformly across layers, though this is not made entirely clear in the writing. Given the variation in sparsity across layers, what is the effect of varying the policy based on the layer?

**Limitations:**

The authors do address limitations

---

> ### Author Rebuttal · Authors · 2023-08-10
>
> We are glad that the reviewer appreciates our work and efforts, and acknowledges we **tackle an important issue** and our approach **has many potentially critical practical implications**. We provide detailed responses below to address the questions.
>
> **[Q1: More Evaluation of Diverse Generation]** Thanks for the suggestion that helps strengthen our observations! We conduct the additional more formal evaluation on the finding of increased diversity by Self-BELU[1] as suggested.
>
> **Result**: The full model reaches a Self-BELU score of **0.0057** while H$_2$O and Local method achieve **0.0051** and **0.0436**, respectively. The comparatively **lower** Self-BELU score of H$_2$O indicates the slightly **increased diversity** of generated text.
>
> **Details**: We randomly sample 100 instances from the XSUM dataset and use them as prompts. Subsequently, we employed the LLaMa-7B model to generate text of equal length.
>
> **[Q2: References and More Observation of Sparsity Finding]**: Thanks for pointing us to the related work that supports our sparsity observations. We have added it to the discussion. Below we present our extended similar sparsity observations in more model families:
>
> **Results**: Figure R1 of the uploaded PDF shows that attention sparsity also exists in LLaMA and GPT-NeoX, where nearly all layers exhibit sparsity levels exceeding **90%**.
>
> **Discussion**: Additionally, we include the missing two references as the following in Section 3.1 in the updated version: “Inspired by previous literature, which reveals the existence of attention sparsity in DistillBERT[3] and bounded-norm self-attention heads[2], we first show an observation that such sparsity of attention also exists in pre-trained LLMs.”
>
> **[Q3: Layer-wise H2O Policy Results]** This is a great suggestion! Indeed, our approach assigns the same KV cache budget for each layer in LLMs. We will clarify this setting in the updated manuscript. Furthermore, we present additional preliminary studies suggested by the reviewer about how the layer-wise cache budget can further help the final performance below.
>
> **Results**: We conducted experiments with three layerwise sparsity-aware strategies. The results in Table R1 suggested that one of them can further boost H$_2$O performance by up to $1.00%$ accuracy and the other two strategies fail and cause a performance degradation ranging from $0.07%$ to $13.20%$. This is a very exciting signal that suggests that if we can find a low-cost algorithm to predict the sparsity level, we can exploit it for a higher compression ratio. We have included the results in our manuscript.
>
> Table R1: Results of layer-wise KV cache budget. Experiments are conducted on LLaMA-7B across different tasks. And the overall KV cache budget is 20%.
>
> ||PiQA|COPA|OpenbookQA|Winogrande|
> |---|---|---|---|---|
> |Full| 80.09 | 81.00 | 44.80 | 71.51|
> |Local|57.94|56.00|28.40|51.30|
> |H2O|79.22|85.00|43.80|71.67|
> |H2O Sparsity-Aware|72.03|78.00|41.20|62.27|
> |H2O Random|72.15|78.00|30.60|64.64|
> |H2O Sparsity-Aware-Custom|79.33|86.00|44.60|72.30|
>
>
>
>
>
>
> **Details**: Three strategies under the same overall KV cache budget (20%): i) we assign layer-wise budget which is proportional to $1-s_i$, where $s_i$ is the attention sparsity levels of layer $i$ (Sparsity-Aware); ii) we first define sensitive and non-sensitive layers based on whether its sparsity ratio $s_i$ is smaller than the model’s average sparsity. Then we assign more KV cache budget to the sensitive layers where we make the budget of sensitive layers $60%$ larger than the ones of non-sensitive layers(Sparsity-Aware-Custom); iii) we assign layer-wise budgets following a uniform distribution (Random).
>
>
> [1] Texygen: A Benchmarking Platform for Text Generation Models
>
> [2] Edelman, B., Goel, S., Kakade, S.M., & Zhang, C. 2022. Inductive Biases and Variable Creation in Self-Attention Mechanisms. ICML.
>
> [3] Likhosherstov, V., Choromanski, K., & Weller, A. 2021. On the Expressive Power of Self-Attention Matrices. ArXiv, abs/2106.03764.
>
> Thanks for reading our response! Please let us know if the above addresses the questions and concerns.

---

> > ### Comment · Reviewer_AZfb · 2023-08-21
> > **Rebuttal Response**
> >
> > Thank you to the authors for taking the time to carefully address my review comments. I think the paper should be accepted.

---

> > > ### Author Response · Authors · 2023-08-21
> > > **Thanks for the responses**
> > >
> > > Dear Reviewer AZfb,
> > >
> > > Thanks for your constructive suggestions and efforts during reviewing. And thanks for your positive evaluations.
> > >
> > > Sincerely,
> > >
> > > Authors

---

### Author Rebuttal · Authors · 2023-08-10

We thank reviewers [AZfb,CXTK,N7jZ,nR7P] for their thoughtful and highly supportive feedback! We were glad that the reviewers found the problem **significant and impactful** [AZfb,CXTK,N7jZ, nR7P], the observations and theory **interesting and well-analyzed** [CXTK, nR7P], the methods **simple and intuitive** [AZfb,CXTK,N7jZ,nR7P], the presentation **easy** to follow [CXTK], experimental results **strong and impressive** [AZfb,CXTK,N7jZ,nR7P].

We have updated the paper to incorporate constructive suggestions, which will show in the revision. We summarize the major changes below:

1. [**nR7P**] **Throughput on Real-World Datasets**:
    - We add  H$_2$O throughput on real-world benchmarks that validate our efficiency claims.
    - We add a discussion to explain why we use simulated settings to assess the throughput improvements, (1) long-form generation benchmarks are rare and still we want to have a wider coverage of system evaluation, (2) our evaluation is independent of existing datasets and could be useful as a unique metric to evaluate future kv cache policies.

2. [**nR7P**] **Stronger Baseline**:
    - We add 0/1 shot as an additional strong baseline, and show that 20%- H$_2$O with 5-shot inputs has up to $9$% better performance across tasks than this baseline.

3. [**AZfb**] **Diversity of Generation Content**:
    - We add a more formal evaluation on the increased diversity by Self-BELU and show that the generated output by  H$_2$O are more diverse than full model, with **lower** self-similarity in terms of BELU (0.0051 vs. 0.0057).

4. [**CXTK, N7jZ**] **Application to Encoder-Decoder / COLT5**:
    - We add H2O results on T5 and show that using **20%** KV Cache can also preserve the performance, similar to auto-regressive cases.
    - We present a discussion on the relation between COLT5 and  H$_2$O, (1) they are different approaches to solve a similar problem (improved efficiency by focusing on important tokens), and conceptually H2O can be applied on the top of COLT5 to further improve its performance; (2) while we do not have access to COLT5, we test  H$_2$O on T5 and show improvement, supports our analysis.

5. [**AZfb**] **Existence of Sparsity and Layer-wise Customization**:
    - We add related work that supports attention sparsity observation and extended similar sparsity (more than 90%) observations on LLaMA and GPTNeo-X.
    - We present preliminary results that layer-wise KV Cache eviction policy according to sparsity can potentially further boost  H$_2$O accuracy by 1 point.

6. [**CXTK, N7jZ**] **Extension to Training**:
    - We add preliminary results for deploying H2O in training on WikiText-103 and show that it could potentially lead to better generalization.

7. [**nR7P**] **Zero / one shot results**:
    - We add zero/one shot results and show that some tasks maintain accuracy under 20% budget and others under 30-40% budget. Generally our  H$_2$O is effective compared to other baselines in this setting.

8. [**CXTK**] **Memorizing transformer**:
    - We add preliminary results on deploying  H$_2$O in memorizing transformers and the positive signals show the possibility of reducing the external memory size of memorizing transformers by introducing heavy-hitter tokens.

---

### Decision · Program_Chairs · 2023-09-21

**Decision:**

Accept (poster)

**Comment:**

The paper addresses the issue of improving the efficiency of LLMs during inference by introducing a novel pruning strategy called H2O. Firstly, the authors present a simple and intuitive approach that is empirically demonstrated to preserve performance across various model architectures and benchmarks in the zero-shot setting. This approach has the potential for significant practical implications in research engineering work involving LLMs. The paper's clear and well-structured presentation is commended, making it easy to follow.

On the other hand, reviewers have highlighted certain weaknesses that need to be addressed. The claim of increased diversity in generations due to H2O is not adequately supported by a small qualitative evaluation and should be reinforced by more rigorous quantitative measures or human studies. Furthermore, the paper should provide citations to related work on sparsity in attention matrices, as the observation of sparsity is not presented as a novel finding. The inconsistency in evaluation between synthetic/toy data and real-world datasets raises concerns about the practical applicability of the proposed method. To address this, the paper should include throughput results on real-world datasets to validate the efficiency claims. Additionally, reviewers have raised questions about the application of H2O during training, its potential use in encoder-decoder frameworks, and its applicability to reducing memory size in Memorizing Transformers.

In response to these comments, the authors have made important revisions to the paper. They have added throughput results on real-world benchmarks, introduced a stronger baseline, conducted formal evaluations of increased diversity, explored the application of H2O in training, and provided preliminary results for its use in Memorizing Transformers. These changes address many of the concerns raised by the reviewers and strengthen the paper's overall contribution.